# Optical Voltage Transformer Based on FBG-PZT for Power Quality Measurement

**DOI:** 10.3390/s21082699

**Published:** 2021-04-12

**Authors:** Marceli N. Gonçalves, Marcelo M. Werneck

**Affiliations:** 1Photonics and Instrumentation Laboratory, Electrical Engineering Program, COPPE—Universidade Federal do Rio de Janeiro, Rio de Janeiro 21949-598, Brazil; mngoncalves@ufrj.br; 2Photonics and Instrumentation Laboratory, Nanotecnology Engineering Program, COPPE—Universidade Federal do Rio de Janeiro, Rio de Janeiro 21949-598, Brazil

**Keywords:** fiber Bragg grating, optical transformer, piezoelectric ceramics, power quality, FBG, OVT, PZT

## Abstract

Optical Current Transformers (OCTs) and Optical Voltage Transformers (OVTs) are an alternative to the conventional transformers for protection and metering purposes with a much smaller footprint and weight. Their advantages were widely discussed in scientific and technical literature and commercial applications based on the well-known Faraday and Pockels effect. However, the literature is still scarce in studies evaluating the use of optical transformers for power quality purposes, an important issue of power system designed to analyze the various phenomena that cause power quality disturbances. In this paper, we constructed a temperature-independent prototype of an optical voltage transformer based on fiber Bragg grating (FBG) and piezoelectric ceramics (PZT), adequate to be used in field surveys at 13.8 kV distribution lines. The OVT was tested under several disturbances defined in IEEE standards that can occur in the electrical power system, especially short-duration voltage variations such as SAG, SWELL, and INTERRUPTION. The results demonstrated that the proposed OVT presents a dynamic response capable of satisfactorily measuring such disturbances and that it can be used as a power quality monitor for a 13.8 kV distribution system. Test on the proposed system concluded that it was capable to reproduce up to the 41st harmonic without significative distortion and impulsive surges up to 2.5 kHz. As an advantage, when compared with conventional systems to monitor power quality, the prototype can be remote-monitored, and therefore, be installed at strategic locations on distribution lines to be monitored kilometers away, without the need to be electrically powered.

## 1. Introduction

Power quality has become an important requirement for electric power systems. The term power quality is applied to a variety of disturbances in electrical systems, which have always been the object of concern for power utilities and industries. The growing use of non-linear electronic loads connected to the network has put the topic even more in focus. Due to its converters, such loads distort the waveform that reaches them, thus “polluting” the network.

On the other hand, the currently used equipment is more sensitive to voltage variations, since many of them have controls based on microprocessors and electronic devices that are sensitive to several types of disturbances, resulting in poor operation and even shortening its useful life.

At the industrial level, such disturbances can affect the continuity of the operation. In some industries, such as steel and petrochemicals, an electrical interruption of up to 1 min can cause significant losses due to the shutdown of machinery, equipment and the production line. For these industries, the economic impacts of energy quality are profound. Additionally, residential consumers have been increasingly informed about the topic of power quality and have demanded that Electric Power Concessionaires improve the service provided. Due to these demands, a set of standards cover many problems associated with power quality [1,2,3]. For instance, the IEEE 1159 recommends the practice for monitoring electric power quality, the IEC 61869-103 describes the use of instrument transformers for power quality measurements, while the IEC 61000-4:30 defines the methods for measurement and interpretation of results for power quality parameters in AC.

For measurement purposes, several models of power quality equipment are available on the market, capable of calculating and storing quality parameters. Although these models are very precise, the challenge is to provide inputs to these monitors with a high degree of accuracy. Power quality equipment can only be used in low voltage. For application in distribution lines, instrument transformers have to be applied to lower the voltage to adequate levels. However, instrument transformers are bulk and heavy, besides needing power to be shut down during installation. Additionally, the power quality meter, connected to the output of the instrument transform, has to be left on the pole to be later collected with the stored information. Because of all these difficulties, power quality is rarely measured in-field, being used only at the substations.

Instrument transformers are widely used in the electric power industry for protection and metering purposes. Instrument transformers can be either voltage transformers (VT) or current transformers (CT), and are responsible of converting high voltage or high current to lower values, compatible to standard low voltage equipment. However, for high voltage applications, voltage and current transformers, when inserted into the measurement chain, could impact the quality of the measured parameters, since they were not devised for quality assessment.

Optical Current Transformers and Optical Voltage Transformers, based on the Faraday and Pockels effect, are a new family of instrument transformers and an alternative to the conventional electric-magnetic transformers for protection and metering purposes. Their advantages are widely discussed in the scientific and technical literature and their commercial applications are well accepted and established [4].

There are only a few manufactures of this equipment in the world, making Faraday and Pockels technologies very expensive. For this reason, many electric plant operators tend to avoid them, not only due to the high costs but also due to the technology relatively new and not dominated by the local technicians, demanding expensive contracts with the manufactures. However, since it has already been proven that optical technology has its merits and advantages, the search for alternative optical technologies is the main objective of high-tech laboratories around the world and also the objective of the present work.

Nevertheless, the literature is still scarce in studies evaluating the use of Optical Transformers for power quality purposes. In this paper, an emerging approach of an OVT based on fiber Bragg grating (FBG) for power quality measurements is presented. The aim is to provide a signal with an appropriate accuracy required by the power quality analyzers and meters.

One of the first studies taking into account OVTs and regulatory standards was presented by Rosolem et al. [5], who carried out a comprehensive review of optical sensors applications for monitoring medium-voltage power quality, discussing technological, economic, regulatory and practical aspects of the installations. In their review, optical transformers based on Pockels and Faraday effects, FBG and Power over Fiber (PoF) techniques were analyzed.

According to Rosolem, to apply optical sensors for monitoring electric power quality, a high precision and a relatively high bandwidth are necessary. Additionally, grid quality monitoring is carried out temporarily and, therefore, the equipment must be easy to install and remove from the electrical distribution network. The sensor must also be cost-effective to justify its use in many electrical distribution nodes [5].

Although several authors have studied the use of piezoelectric materials, such as PZT (lead zirconate titanate—PbZrTi or PZT for short) and FBG for voltage measurements, in general, the tests are performed for AC voltages without any type of disturbance and with a fixed frequency (50 or 60 Hz). OVT’s response to possible voltage disturbances that compromise the quality of the grid is fundamental for application in measurement systems.

Ribeiro et al. [6] tested their sensor at AC voltages up to 2 kV at 60 Hz. The supply voltage was generated by an AC voltage source. Gonçalves and Werneck [7] used the signal from the network itself connected to a variac and then to a high voltage transformer, while Fusiek et al. [8] used an AC source connected to a high voltage transformer to generate the signals to the sensor.

Nasir et al. [9] evaluated the possibility of using a hybrid sensor for voltage and current measurements to detect faults in the electrical power system. The sensor’s response to a fault occurrence was simulated considering its theoretical model, but voltage and current responses under faults were not applied to the sensor.

Dante [10] exposed his PZT-based sensor to a voltage transient, applying a 60 Hz sinusoidal signal modulated by a 1 Hz square wave. For this purpose, a signal generator was used, connected to a preamplifier coupled to an elevator transformer. An abrupt transient in the amplitude of the high voltage signal was applied (1.77 kV_rms_ to 2.66 kV_rms_) and it was possible to observe the optical sensor following the voltage variation, presenting only a phase shift characteristic of the experimental set-up.

Yang et al. [11] tested their setup with sinusoidal, square and triangular input signals at frequencies of 50 Hz, 3 kHz and 8 kHz. To generate the input signals, a signal generator connected to an amplifier was used. The maximum applied voltage was 2.5 kV. As a result of this study, strong consistency was observed between the input voltage for different waveforms and the sensor responses at different frequencies. Distortions were observed mainly for triangular and square wave inputs. For the sinusoidal input, the output signal was distorted when the applied frequency was close to the resonance and antiresonance frequencies of the PZT material used for measuring the electric field.

Recently, Dante et al. [12] reproduced his test applying a higher voltage transient (zero to 10 kV_rms_) to emulate a step response. Once more, the sensor reproduced the input voltage variation with a response time of the output signal less than 1 ms. Nevertheless, the author points out that the observed delay is mostly due to the low-pass filters implemented in the electronics of the interrogation system.

Finally, the work of Fusiek and Niewczas [13] seems to be the first effort to test a photonic voltage transducer under an official and generally accepted IEC standard.

Although the above-mentioned works applied voltage variations on their sensors, none of them set out to carefully study the sensor’s behavior under different disturbances. Additionally, all previously mentioned works developed their sensors in the laboratory without concerning their use in-field.

This paper describes the development of a temperature-independent prototype of an optical voltage transformer based on FBG-PZT, adequate to be used in field surveys. Then, the OVT was tested under several disturbances defined in IEEE standards that can occur in the electrical power system, especially short-duration voltage variations. The results demonstrated that the proposed OVT presents a dynamic response capable of satisfactorily measuring such disturbances and that it can be used as a power quality monitor for a 13.8 kV distribution system. The prototype is temperature-independent applying a passive thermal compensation technique previously presented by Gonçalves and Werneck [7]. Additionally, due to fact that FBGs can be remote-monitored without loss of information, as demonstrated in [14], the proposed system can be installed at specific locations on distribution lines to be monitored kilometers away, without the need to be electrically powered.

## 2. Materials and Methods

### 2.1. OVT Working Principle

This section starts with a presentation of the theory of FBG sensors and PZT ceramics, fundamental for understanding the working principle of the proposed optical voltage transformer. In the sequence, we show the principle of operation of the sensor for power quality measurements.

The Bragg grating consists of a periodic modulation of the refractive index of the fiber core. These modulations are called gratings and, in their simplest form, are perpendicular to the axis of light propagation. When a broad band light introduced into the fiber, and reaches the grating, a narrow portion of the spectrum will be reflected by the periodic variation of the refractive index, due to the interaction with different refractive indices in its core. The reflected light signal satisfies the Bragg condition, given by [15]:(1)$$\lambda_{B}=\;2\;n_{\mathrm{eff}}\;\Lambda ,$$
where n_eff_ is the effective index of refraction of the fiber core, Λ is the modulation period of the index of refraction and λ_B_ represents the peak of the reflected spectra, known as the Bragg wavelength.

A longitudinal deformation, due to an external force, can alter both n_eff_ and Λ, through the photoelastic effect. In addition, a temperature variation is also able to change the two parameters, as a result of the thermo-optical effect and the thermal expansion of the silica. Thus, the FBG is essentially a temperature and strain sensor. However, several parameters can be measured if correctly related to the parameters that directly affect the Bragg wavelength.

The Bragg Equation (2) establishes the relationship between the Bragg wavelength, the strain and the temperature applied to an FBG [16]:(2)$$\frac{\Delta \lambda_{B}}{\lambda_{B}}\;=\;\left(1-\rho_{e}\right)\varepsilon_{\mathrm{FBG}}\;+\;\left({\eta \;+\;\alpha }_{\mathrm{FBG}}\right)\Delta T,$$
where $\varepsilon_{\mathrm{FBG}}$ is the strain of the fiber (ΔL_FBG_/L_FBG_), ΔT is the temperature variation, $\rho_{e}$ is the photo-elastic coefficient of silica (0.22), η is the thermo-optical coefficient (8.6 × 10^−6^ °C^−1^) and $\alpha_{\mathrm{FBG}}$ is the coefficient of thermal expansion of the silica (0.85 × 10^−6^ °C^−1^).

For an optical fiber with an FBG centered at 1550 nm, it is possible to obtain the FBG theoretical sensitivity for strain and temperature by substituting data from Table 1 in Equation (2) leading to Equations (3) and (4). Those are theoretical values, but by calibrating the FBG to strain and temperature, we get to approximately the same values:(3)$$\frac{\Delta \lambda_{B}}{\varepsilon_{\mathrm{FBG}}}=1.21\;\mathrm{pm}/\mu \varepsilon$$
(4)$$\frac{\Delta \lambda_{B}}{\Delta T}\;=\;14.18\;\mathrm{pm}/{}^{\circ}C$$

The piezoelectric effect can be defined as the conversion of mechanical to electrical energy (named the direct piezoelectric effect) or the conversion of electrical to mechanical energy (named the reverse piezoelectric effect).

Some crystals, such as quartz, present a natural piezoelectric property, but the piezoelectric effect can also be obtained in some ceramics such as PZT.

When a PZT is subjected to an electric field, its dimensions will change according to the reverse piezoelectric effect. The relation between the strain (ε) of the piezoelectric material and the electric field (E) applied into this material is determined by the piezoelectric charge constant (d_ij_):(5)$$d_{\mathrm{ij}}\;=\;\left[\;\frac{\varepsilon }{E}\;\right]{\mathrm{mV}}^{-1}$$

The constant d_33_ defines the parallel deformation to the polarization vector of the ceramic, in its width. For a ceramic ring, with electrodes deposited on its faces, its width (w_o_) will vary according to:(6)$$\Delta w\;=\;E\;d_{33}w_{o},$$
where E is the electric field and can be represented as V_in_/w_o_.

In this case, V_in_ is the input voltage applied between the PZT electrodes, in volts, and w_o_ is the ring thickness. Hence, the variation in the ceramic width could be rewritten as:(7)$$\Delta w\;=\;V_{\mathrm{in}}\;d_{33}$$

Since the strain that occurs in the PZT is proportional to the applied voltage, an FBG fixed to the PZT ends can be used to gauge the voltage. This technique is applied in most of the works mentioned so far. However, PZTs, as all ceramics, present a thermal dilation coefficient that will also strain the FBG, and thus, compete with the small strain due to the electric field being measured. For this reason, the mechanical assembly was devised to compensate the thermal dilation of all materials used, as will be described below.

While applying AC voltages to the piezoelectric material, it will compress and expand at the same frequency, as per Equation (7). The FBG fixed on the PZT will follow the movement of its thickness, producing a proportional variation in the Bragg wavelength.

When using ring-shaped ceramics, the FBG could also be fixed inside the ceramic cavity. To increase the sensitivity of the set, stacked ceramics are often used [6,7,8]. In this configuration, for the FBG to follow the displacement of the entire stack, different approaches of assembling the stack must be evaluated.

Nevertheless, in all the configurations mentioned, the principle of operation of this type of sensor is based on the premise that the strain of the FBG will be proportional to the displacement of the PZTs ceramics, which in turn is proportional to the voltage applied between the PZT electrodes, and thus, it is possible to obtain a relationship of the Bragg wavelength with the applied voltage: ∆λ_B_⁄∆V_in_.

### 2.2. OVT Mechanical Structure and Sensitivity

A drawing of the mechanical structure of the Optical Voltage Transformer is shown schematically in Figure 1. The assembly includes a stack of 10 PZT rings fixed between two aluminum sustention plates (one is fixed, while the other is movable); two coaxial screws, each one fixed to one sustention plate; 11 copper electrodes inserted between the PZT rings; and an FBG bonded between the screw extremities. The screws are coaxially arranged to allow one sliding inside the other, and each one fixed to one sustention plate. Therefore, any elongation or shrinking of the PZT stack will be transmitted through the screws to the FBG.

Figure 2 shows a picture of the OVT assembly with all components.

Since the whole system is subjected to temperature variations, it is expected that the FBG will be strained by the different thermal dilation of each material used in the assembly. Therefore, to compensate for temperature fluctuations, all materials used in the assembly were carefully chosen so as to guarantee that the distance between the two screw ends to which the FBG is bonded will not change for the full temperature range. Figure 3 shows all materials involved in the assembly.

For this calculation, we defined a fixed reference (hatched line) at the left of the illustration, located at the outer side of the left sustention plate.

The upper series is comprised of the sustention plate, the PZT rings, all electrodes and the outer screw, whereas the lower series is made of the inner screw and the FBG bonded between the inner and outer screw. The outer screw is attached to the movable sustention plate and the inner screw is attached to the fixed sustention plate so that both ends lie outside the structure to allow the FBG to be bonded between them (see Figure 1). When the PZT rings expand or contract due to the electric field applied, the assembly tends to move together except the fixed sustentation plate, therefore moving the screws, one inside the other. The result of this movement is that the screw ends will displace the same as the PZTs. Now, if the upper series dilates exactly the same as the lower series, the FBG will not sense any stress due to temperature. For accomplishing this idea, the inner screw was made of carbon steel and the outer screw was made of invar both with precise a length. A detailed methodology on how to achieve thermal stabilization was described in [7].

This strategy allows a better sensitivity since the PZT rings are serially connected in a stack but electrically connected in parallel, so as to multiply the displacement by 10, and do not increase the necessary voltage to displace the rings.

Table 1 shows the properties of the FBG and PZT used on the OVT.

The PZT-4, manufactured by Sparkle Ceramics (Essex, UK), was chosen to compose the set. This PZT is known as hard type and is recommended for medium and high-voltage applications. It has a low piezoelectric charge constant and a high resonance frequency, which guarantees that the set will not resonate close to the normal working frequency of 60 Hz. However, this frequency limits the number of voltage disturbances that the OVT can measure.

When a voltage is applied between the PZT electrodes, all the PZT rings will expand their thickness, as shown in Equation (7). The displacement will be transferred to the FBG through the screws where the FBG is bonded. Then, we have:(8)     ΔLFBG=Δwor          ΔLFBG=N ΔVin d33 ,
where ΔL_FBG_ is the displacement of the FBG and N is the number of ceramic rings.

Considering the temperature compensated, we obtain the sensitivity of the OVT under voltage variation by substituting Equation (8) in Equation (2):(9)$$\frac{\Delta \lambda_{B}}{\Delta V_{\mathrm{in}}}=\left(1-\rho_{e}\right)\frac{N\;d_{33}}{L_{\mathrm{FBG}}}\;\lambda_{B}$$

The value of the sensitivity could be obtained by substituting the parameters of Table 1 in Equation (9):(10)$$\frac{\Delta \lambda_{B}}{\Delta V_{\mathrm{in}}}=232\;\mathrm{pm}/\mathrm{kV}$$

This result means that for each kV applied on the OVT electrodes, a proportional displacement of 232 pm will occur in the FBG’s Bragg wavelength (λ_B_).

Better sensitivities can be achieved by choosing a different PZT type with a higher piezoelectric charge constant, for instance, PZT-5A or PZT-5H. The number of elements in the PZT stack also influences the increased sensitivity; however, this should be increased with caution, since the assembly could become heavy and bulky. Another way of enhancing the sensitivity is to decrease the FBG length, and reducing the distance between the end of inner and outer screws, where the FBG is bonded, since this would increase the strain applied to the FBG.

### 2.3. Interrogation System

An interrogation method is necessary to measure the Bragg wavelength displacement. Commercial interrogators are in general expensive equipment, especially for high frequencies, which is the case for applications for power quality measurements. For this reason, we designed our interrogation system based on a manually adjustable Fabry-Perot (F-P) tunable filter (Model DTS0051, OZ Optics Ltd, Ottawa, ON, Canada), which showed to be a good alternative to commercial interrogators since it allows high-frequency interrogation at a lower cost. Figure 4 shows the interrogation setup used for demodulating the FBG signal, where an F-P tunable filter is applied. The demodulation technique consists of converting the frequency modulated signal into an amplitude modulated signal by an edge filter.

The numbered squares in Figure 4 represent the spectrum of the signal as it travels through the different components of the demodulation setup. The broadband light source (Model ASE730, Thorlabs, Newton, NJ, USA) generates a large-spectrum signal on the telecom Band C (1). The signal crosses the circulator (2) and illuminates the FBG (made in our laboratory by Nd-YAG laser), which reflects its Bragg wavelength (3). The Bragg wavelength reaches the Fabry-Perot filter, which is tuned to the center of the trailing slope of the FBG reflected spectrum (4). The effect of this filtering is a convolution of both spectra, represented by their superposition shown in blue (5), which is directed to the photodetector. As the PZT expands and contracts due to the applied electric field, the convolution area changes, feeding a variable optical power into the photodiode, therefore converting the FBG displacement peak into an amplitude modulated signal that is seen at the oscilloscope.

Note that this technique does not apply to temperature-drifted signals since the displacement due to temperature would detune the process and change the relationship between input frequency to output power. However, since the OVT is temperature stabilized, no changes occur in the convolution process. The amplitude-modulated signal resulting from the convolution is directed to a photodetector which produces an electrical signal proportional to the Bragg wavelength displacement. Then, the signal can be amplified, filtered, and visualized on an oscilloscope or computer, depending on the application, as shown in the diagram in Figure 4.

### 2.4. OVT Output Voltage

A commercial photodetector coupled to a transimpedance amplifier (Model PDA10CS, Thorlabs, Newton, NJ, USA) was used to provide a voltage signal proportional to the measured optical signal.

The output voltage of the transimpedance amplifier is given by:(11)$$\Delta V_{\mathrm{out}}=A_{R}\;I_{\mathrm{PD}},$$
where A_R_ is the transimpedance gain of the amplifier given by the datasheet: 4.75 × 10^6^ V/A when adjusted to 70 dB and I_PD_ is the current in the photodetector.

The current in the photodetector can be obtained through the input power of the optical signal, as shown by Equation (12):(12)$$I_{\mathrm{PD}}=R\left(\lambda \right)P_{\lambda },$$
where $R\left(\lambda \right)$ is the photodetector responsivity equal to 1.1 A/W according to the photodetector datasheet and $P_{\lambda }$ is the optical power incident at a given wavelength.

Finally, the photodetector input power is given by:(13)$$P_{\lambda }=k\;\Delta \lambda_{B},$$
where k is the sensitivity resulting from the convolution between the reflectance spectrum of the FBG and the transmittance spectrum of the Fabry-Perot filter, equal to 4.3 × 10^−8^ W/nm.

Combining Equations (10)–(13) and substituting the constants, we obtain the ratio between the output and input voltage of the optical transformer:(14)$$\frac{\Delta V_{\mathrm{out}}}{\Delta V_{\mathrm{in}}}=52.12\;\mathrm{mV}/\mathrm{kV}$$

### 2.5. Experimental Set-Up for Quality Measurement

The OVT was submitted to different disturbances to evaluate its response. Figure 5 shows the experimental setup used to apply the generated disturbances to the OVT and analyze its performance.

The Simulink module of MATLAB was chosen as the software for modeling and simulating the disturbances to be input and measured by the OVT. The simulated signals were storage and inserted in a signal generator (Model AFG1022, Tektronix Inc., Beaverton, OR, USA), capable to properly reproduce the disturbances. As the generator output is a low voltage and low power signal, the signal is first directed to a signal amplifier (Model WVOX A2000, Machine Audimax, São Paulo, Brazil, 360 W output power, 20 Hz to 20 kHz frequency range, harmonic distortion <0.05%), which provides the proper input signal to an elevator transformer with a transformation ratio of 23,000:127 V/V. The elevator transformer is a conventional instrument transformer for 13.8 kV distribution lines.

The transformer is responsible for generating voltages up to 4 kV, which is the maximum voltage supported by the PZT stack. In a real-life application, this system is supposed to be used in a 13.8 kV distribution line, which provides about 8 kV from phase to ground potential (13.8 kV/√3). Since the maximum supported voltage of the PZT stack is 4 kV, a capacitive divisor will be employed, as will be discussed below.

An oscilloscope (Model DS1102CA, Rigol, Suzhou, China) with a high-voltage probe with an attenuation of 1:1000 (Model HV40A, Minipa, São Paulo, Brazil) was used to measure the input signal applied to the OVT. The set with the elevator transformer, the OVT and the high voltage probe was isolated into a high voltage room in order to guarantee the protection of the experiment and operators. The optical fiber from the FBG brings to the interrogation system (see Figure 4) the OVT response to the applied disturbances.

## 3. Results and Discussion

To validate the behavior of the transformer under different disturbances, short-duration voltage variations were chosen, since they are the most common disturbances in the medium-voltage electrical power system. The advantage to use Simulink as simulation software is that all types of disturbances can be simulated and directed to the optical transformer through the scheme of Figure 5.

Three different phenomena were simulated: sag, swell and interruption. Figure 6 shows the model created in MATLAB/Simulink to simulate the disturbances and Figure 7 shows the type of fault for each one. Voltage sag is caused by a phase-to-earth fault of a 13.8 kV distribution line. Single-phase faults generate low voltage sags; however, they are unbalanced and asymmetric. Voltage swell is caused by a phase-to-phase fault with a high fault resistance, while the interruption is caused by a phase-to-phase-to-earth short circuit.

Figure 8 shows the signal generator output voltage (CH1) and the signal amplifier output voltage (CH2) for each disturbance. They were applied for three cycles, which is equivalent to a short-duration voltage variation known as instantaneous for sag and swell, ranging from 0.5 to 30 cycles, and momentary for interruptions ranging from 0.5 cycles to 3 s, according to the standard IEEE 1159 (see Table 2). The adequate reproduction of the input signal to the output by the amplifier is guaranteed by its frequency range, from 20 Hz to 20 kHz, and a harmonic distortion <0.05%.

The amplified signals shown in Figure 8 were directed to the instrument transformer located in the high-voltage room. Figure 9 shows the optical voltage transformer, immersed in insulation oil. The other components of high voltage room are the elevator transformer beneath the bench, the high voltage probe and the optical fiber directed to the interrogation system. Another high voltage probe connected to a multimeter was required to adjust the working voltage to 4 kV_RMS_. The probe requires a voltmeter with an input impedance 10 times higher than its output impedance to properly show the voltage divided by 1000. The oscilloscope, on the other hand, has an input impedance in the same order of magnitude of the probe. Due to this fact, the value displayed in the oscilloscope is about half the RMS value applied to the OVT.

The capacitive divider on the right side of the bench is used when testing the OVT for the higher voltage values found on 13.8 kV distribution lines.

### 3.1. Short-Duration Variation Analysis

The first experiment was realized by applying an instantaneous sag disturbance for three cycles of 60 Hz (50 ms) in the OVT. The result is presented in Figure 10, where CH1 is the input voltage in the OVT and CH2 is its output voltage after demodulating the optical signal. It is possible to notice that the output signal properly reproduces the phenomena introduced in the input. To check for reproducibility, we calculated the input-output ratio. Since the real input voltage as measured by the high voltage probe presents an attenuation of 1:1000 and it is twice the displayed value due to the input impedance of the oscilloscope, the ratio between the output and input voltage is equal to 50.54 mV/kV. The relative error is 3.03% concerning the theoretical value in Equation (14). The phase displacement is equal to 8.63°.

The second experiment explores the instantaneous swell disturbance (Figure 11) and the final experiment tests the OVT over a momentary interruption (Figure 12). Both results show the output voltage reproducing the power quality phenomena of the input voltage presenting a ratio between the output and input voltage equal to 50 mV/kV. The relative errors of both swell and interruption are 4.07% and the phase displacement is the same as in the first experiment.

The other method to check adequate reproducibility is to measure the Total Harmonic Distortion (THD) of the output and compare it with the THD of the input. For this calculation, it is necessary to take the Fast Fourier Transform (FFT) of both the input and output of the three disturbances. The FFTs of the disturbances are shown in Figure 13 up to the 10th harmonic.

Note that most of the power of the FFTs is concentrated on the three first harmonics, which was expected, as all three disturbances mostly contain 60 Hz sinusoidal waveshapes. Nevertheless, further analysis was necessary to check for possible high-frequency distortions.

From the harmonic components of the FFT, we can obtain the THD of input and output, as well as the difference between them, representing the distortion introduced by the proposed OVT, as shown in Table 3.

The obtained results demonstrate that optical voltage transformer is capable of reproducing short-duration voltage variations applied to its input with a relative error smaller than 5% and introducing a THD smaller than 2%.

### 3.2. Transient Analysis

To validate the behavior of the OVT under transients (see Table 2), a simulation of an erroneous start of a capacitor bank was generated in MATLAB, generating an impulsive transient as shown in Figure 14a. However, after the signal was directed to the instrument transformer, it was suppressed, since the transformer is manufactured for working with a nominal frequency of 60 Hz and its equivalent impedance is composed of inductance and resistance, it works as a low-pass filter and its response to frequencies above its cut-off frequency will be attenuated. Figure 14 shows the OVT input voltage with an impulsive transient (a) and its output voltage (b) in which the transient was suppressed. This test concludes that the transformer does not respond to transients with frequencies above 1 kHz, and therefore, another method has to be used to test the OVT behavior under high frequencies.

A second test to evaluate the sensor’s behavior under high frequencies was performed by simulating a step function in the signal generator. This signal was injected into the amplifier (Figure 15a, CH1) and the amplifier output signal (Figure 15a, CH2) was injected into the transformer. However, the instrument transformer also suppressed the amplified step, transforming it into an attenuated rising exponential (Figure 15b, CH1). The OVT responded with a similar signal (Figure 15b, CH2).

Since it was not possible to reproduce high-frequency disturbances by the instrument transformer such as transient and step, we decided to use sinusoidal waveshapes of several frequencies to test the highest frequency the elevator transformer would respond to. By increasing the input frequency of the sinusoidal waveform to the transformer, we observed that at 1 kHz its output signal was about 536 V to an input voltage of 4000 V, which means that its 3 dB cut-off frequency is far below this frequency (Figure 16).

It is possible to calculate the frequency response of the instrument transformer by measuring the rise-time of its response to a step function (Figure 15a). The rise-time is 5.2 ms and by applying the known low-pass filter equation:(15)$$\tau =\frac{0.35}{f_{3\mathrm{dB}}},$$
where *τ* is the rise-time and f_3dB_ is the cut-off frequency, we calculated the transformer cut-off frequency to be 67.3 Hz. This means that any frequency above 67.3 Hz will be attenuated at a ratio of −20 dB/decade.

The same procedure could be applied to the OVT to calculate its frequency response but, as mentioned above, no step function can be generated by the transformer. However, another approach can be applied to know which frequencies can be transmitted by the OVT. The approach is to calculate the Fast Fourier Transform (FFT) of an input signal applied to the OVT and compare it with the FFT of its corresponding response. Then, for each frequency pair (input and output), we can compare their amplitudes to check when the attenuation reaches the −3 dB cut-off frequency. Figure 17 shows the two FFTs taken from the signals shown in Figure 15b: the input and output of the OVT.

Figure 17 (right) shows input FFT superimposed with the output FFT for frequencies up to 2.5 kHz. Note that they are very close to each other, but to reveal small differences, Figure 17 (left) presents an amplified portion of the superimposed FFTs. Is possible to see that in some frequencies, the OVT response slightly attenuates the input signal and in some other frequencies the opposite happens.

Figure 17 (right) also shows the OVT’s response in decibel (dB) (yellow line). In all frequencies up to 2.5 kHz, the response does not reach the −3 dB point, which means that the cut-off frequency lies above 2.5 kHz. The conclusion is that, although the cut-off frequency was not detectable, the OVT is capable to reproduce at least up to the 41st harmonic without distortion

Since the OVT was capable to reproduce up to 2.5 kHz without distortion, it is possible to conclude that it is capable to reproduce at least a 400 μs impulsive transient and a low-frequency oscillatory transient, increasing the number of power quality disturbances that the proposed OVT is capable to measure.

## 4. Field Prototype Proposal

The OVT presented in this paper was designed to be installed in-field in order to detect localized disturbances on the 13.8 kV distribution line. For this purpose, the OVT must be capable to withstand the line voltage of each phase, that is, 8 kV, between any phase and ground potential. Since the maximum supported voltage of the PZT stack is 4 kV, a capacitive divisor was employed.

In high and extra high voltage transmission systems, capacitor voltage transformers (CVTs) are used to provide potential outputs to metering instruments and protective relays. In CVTs, such as the one shown in Figure 18a, a capacitive divisor is used to lower the high voltage and apply it to a medium voltage inductive transformer, which will provide in its output a low-voltage sample of the high voltage being measured. The basic circuit diagram for a typical CVT is shown in Figure 18b. It is desirable that the CVT operates as close as possible to an ideal voltage divider, but the induction of the transformer produces a phase delay. The compensating inductor L is designed to correct the voltage lag.

In the case of the proposed OVT, it is also necessary to lower the input voltage, but as the PZT stack has its own capacitance (see Table 1), the capacitor C2 and inductor L are not necessary; it is just a case to add another capacitor in series to perform a divisor. This capacitor, however, has to withstand 4 kV in order to allow another 4 kV to be applied to the OVT. To create such a high voltage capacitor, it is necessary to connect several 400 V rated capacitor in series, as shown in Figure 19 (Left). Then, the assembly is housed inside a 13.8 kV bushing, whereas the OVT is protected inside an IP65-rated casing, as shown in the illustration of Figure 19 (right).

## 5. Conclusions

In this paper, a novel application proposal for an optical voltage transformer based on fiber Bragg grating and piezoelectric ceramics is presented. In this proposal, for the first time, an optical voltage transformer was tested under different disturbances presented in the standard IEEE 1159-2019. The focus of the experimental tests was the short-duration voltage variation, since they are the most common disturbances in medium voltage lines. The OVT was capable to reproduce in its output a sag, swell and interruption disturbances. In addition, it was concluded that the proposed optical voltage transformer was capable to reproduce up to the 41st harmonic without significative distortion and impulsive surges up to 2.5 kHz.

The paper also presented a field proposal for 13.8 kV distribution line, where a capacitive divisor is employed to reduce the voltage to acceptable levels by piezoelectric ceramics used in the mechanical structure of the transformer. Since the OVT output is an optical signal, the field prototype could be installed on any distribution line and be monitored from kilometers away. After going through an interrogation system, the final output signal is a voltage signal that could be directed to power quality analyzers and meters.

The main contribution of this paper was to demonstrate an FBG-based OVT capable to be used in power quality that can be installed at specific strategic locations and be monitored by a simple method, possibly making it a piece of competitive equipment.

The obtained results demonstrate that the proposed OVT is capable to reproduce short-duration voltage variations applied to its input with a relative error smaller than 5% and introducing a THD smaller than 2%.

The proposed system can also be applied on higher voltage lines, such as 138 kV, 230 kV or 500 kV, just by adding a suitable capacitive divisor.

As a suggestion of future studies, such a system shall be installed in a 13.8 kV distribution line in parallel with a conventional power quality system to compare their output measurements.

## Figures and Tables

**Figure 1 sensors-21-02699-f001:**
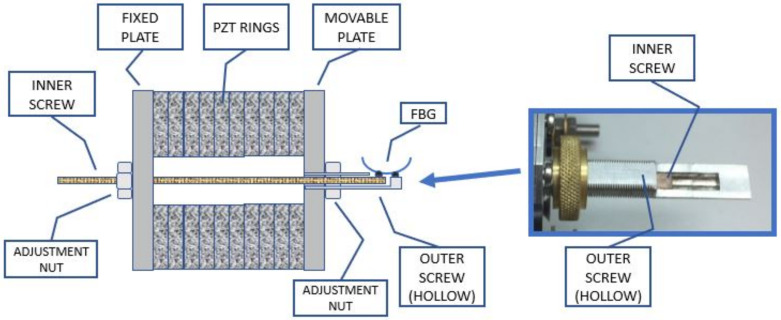
Method used to apply an FBG to monitor the PZT ring stack. The box on the right shows an enlarged picture of the tips of the screws, between which the FBG was bonded (FBG not show).

**Figure 2 sensors-21-02699-f002:**
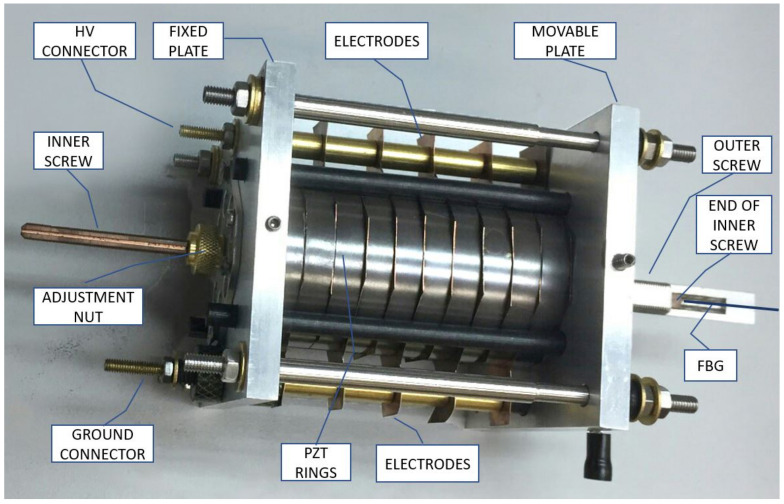
OVT mechanical structure with PZT ceramics stack. The FBG is bonded between the screw ends (adapted from [7] with permission).

**Figure 3 sensors-21-02699-f003:**
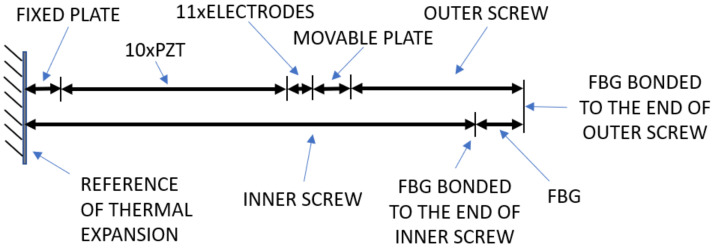
Diagram of the temperature compensation scheme. By choosing the material and length of the two screws, the distance between their ends is independent of the temperature (adapted from [7] with permission).

**Figure 4 sensors-21-02699-f004:**
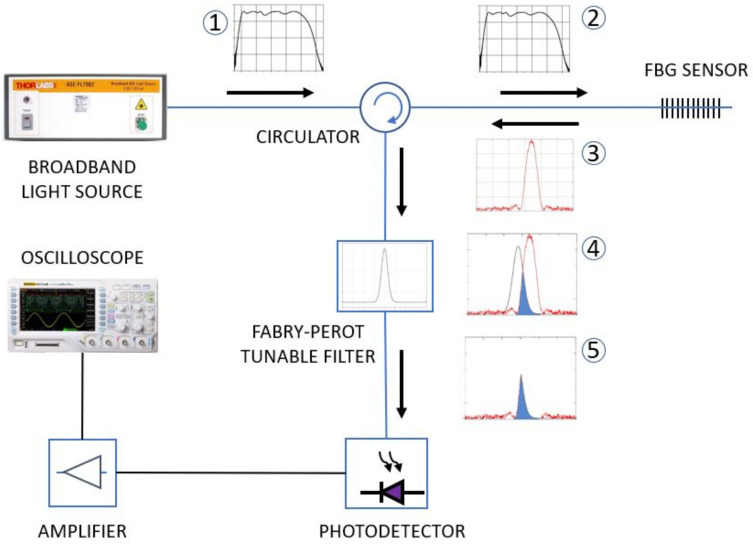
The FBG interrogation setup based on a Fabry-Perot tuned filter. Numbered squares show the spectra of the signal as it travels through the different components until reaching the photodetector, where it is converted into a voltage proportional to the FBG peak displacement and read by the oscilloscope.

**Figure 5 sensors-21-02699-f005:**
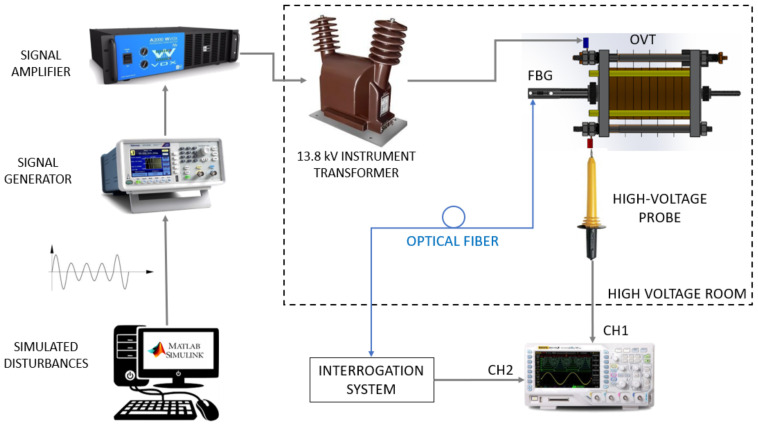
Experimental set-up for power quality measurements. Dotted box represents the high voltage cubicle. The interrogation system is show in Figure 4.

**Figure 6 sensors-21-02699-f006:**
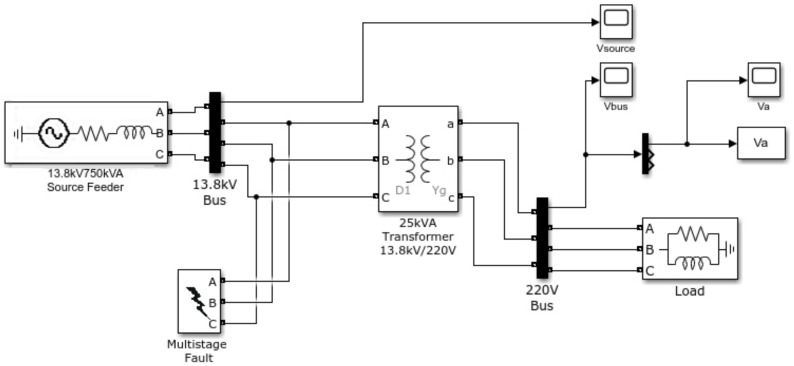
MATLAB/Simulink model for simulated short-duration voltage variations.

**Figure 7 sensors-21-02699-f007:**
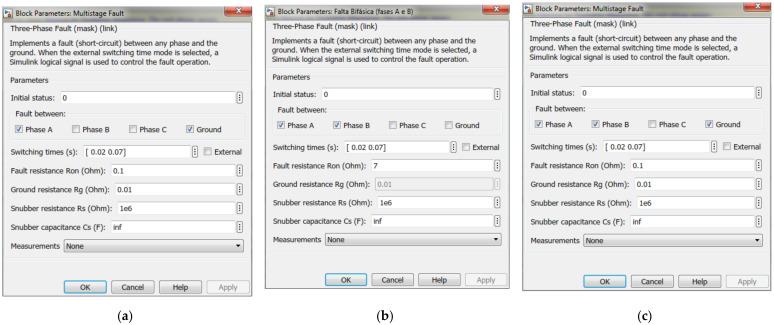
Fault types. (**a**) Phase-to-earth causes voltage sag, (**b**) phase-phase causes voltage swell and (**c**) phase-phase-to-earth causes interruption.

**Figure 8 sensors-21-02699-f008:**
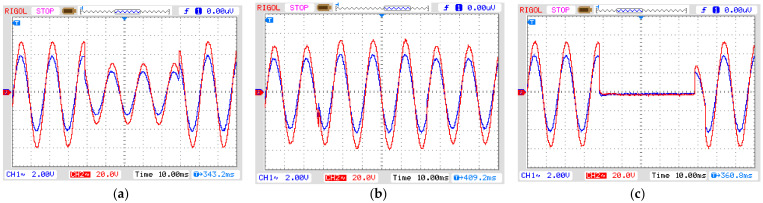
Simulated short-duration voltage as seen from input and output of the amplifier. (**a**) Sag, (**b**) swell and (**c**) interruption. Blue line: input signal from signal generator to amplifier; Red line: output from amplifier (see Figure 5). In all graphs: vertical axis with 2 V/div and horizontal axis with 10 ns/div.

**Figure 9 sensors-21-02699-f009:**
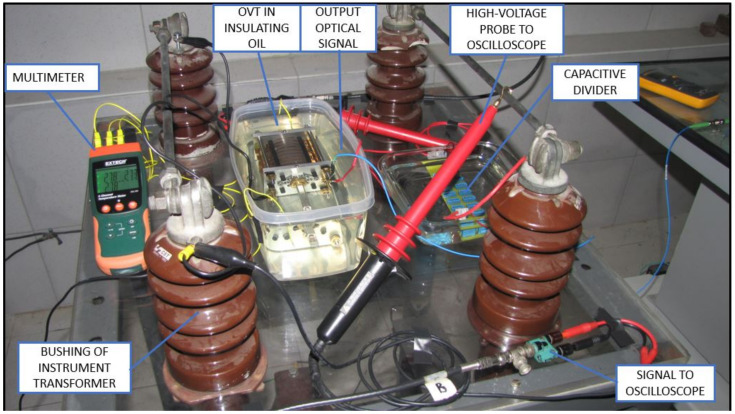
Experiment setup in high voltage room (adapted from [7] with permission).

**Figure 10 sensors-21-02699-f010:**
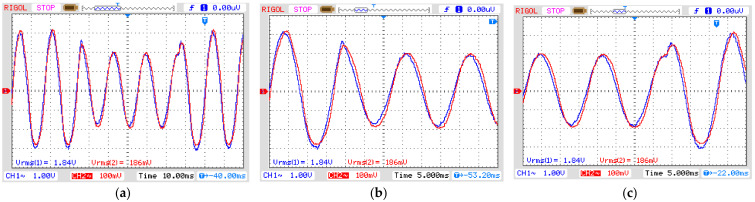
OVT input (CH1) and output (CH2) voltage for sag disturbance. Vertical axis: 1 V/div (CH1) and 100 mV/div (CH2). (**a**) Complete disturbance (horizontal axis: 10 ms/div); (**b**) Beginning of disturbance (horizontal axis: 5 ms/div); (**c**) End of disturbance (horizontal axis: 5 ms/div).

**Figure 11 sensors-21-02699-f011:**
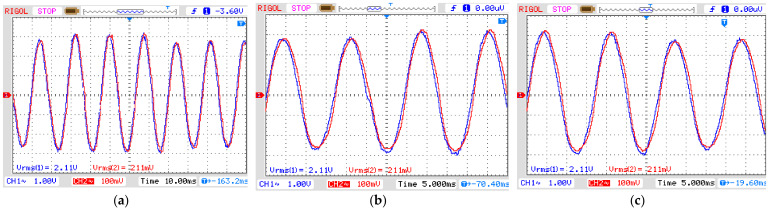
OVT input (CH1) and output (CH2) voltage for SWELL disturbance. Vertical axis: 1 V/div (CH1) and 100 mV/div (CH2). (**a**) Complete disturbance (horizontal axis: 10 ms/div); (**b**) Beginning of disturbance (horizontal axis: 5 ms/div); (**c**) End of disturbance (horizontal axis: 5 ms/div).

**Figure 12 sensors-21-02699-f012:**
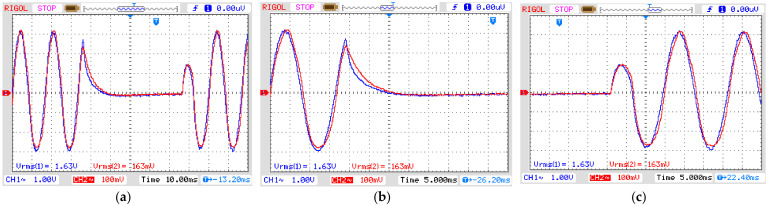
OVT input (CH1) and output (CH2) voltage for INTERRUPTION. Vertical axis: 1 V/div (CH1) and 100 mV/div (CH2). (**a**) Complete disturbance (horizontal axis: 10 ms/div); (**b**) Beginning of disturbance (horizontal axis: 5 ms/div); (**c**) End of disturbance (horizontal axis: 5 ms/div).

**Figure 13 sensors-21-02699-f013:**
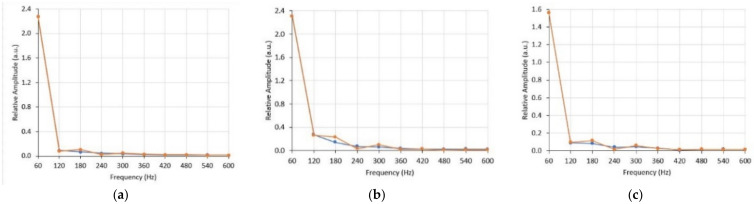
FFT of sag (**a**), swell (**b**) and interruption (**c**) disturbances, with the input (blue line) and output (red line) superimposed.

**Figure 14 sensors-21-02699-f014:**
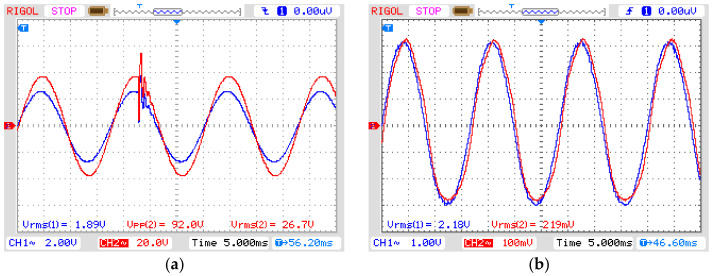
Impulsive transient disturbance. (**a**) CH1: input signal from signal generator to amplifier, CH2: amplifier output voltage (vertical axis: 2 V/div (CH1) and 20 V/div (CH2), horizontal axis: 5 ms/div); (**b**) CH1: OVT input voltage, CH2: OVT output voltage (vertical axis: 1 V/div (CH1) and 100 mV/div (CH2), horizontal axis: 5 ms/div).

**Figure 15 sensors-21-02699-f015:**
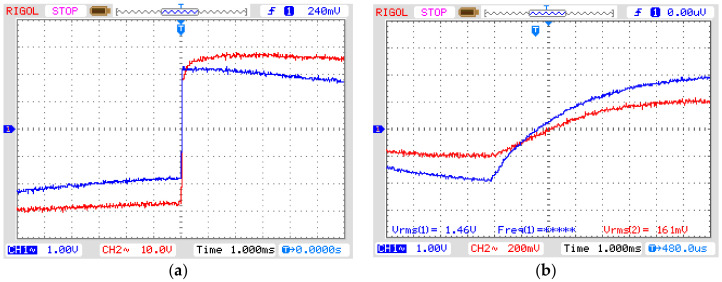
Step response. (**a**) CH1: input signal from signal generator to amplifier, CH2: amplifier output voltage. Vertical axis: 1 V/div (CH1) and 10 V/div (CH2). Horizontal axis: 1 ms/div; (**b**) CH1: OVT input voltage, CH2: OVT output voltage. Vertical axis: 1 V/div (CH1) and 200 mV/div (CH2). Horizontal axis: 1 ms/div.

**Figure 16 sensors-21-02699-f016:**
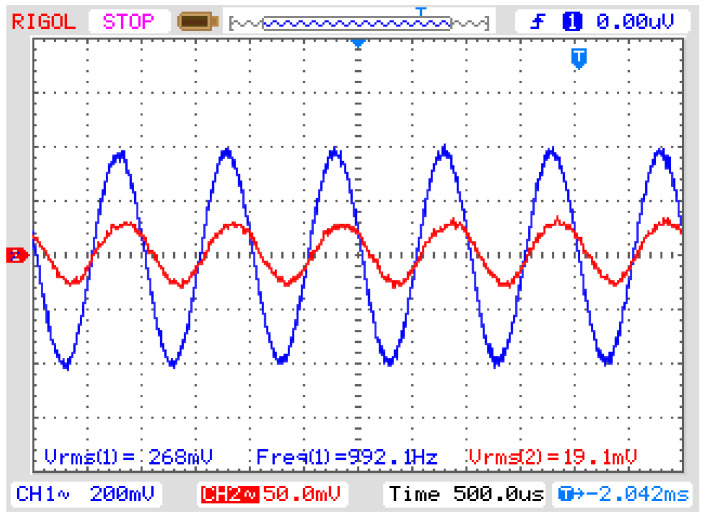
OVT input (CH1) and output (CH2) voltage for a 1 kHz sine wave. Vertical axis: 200 mV/div (CH1) and 50 mV/div (CH2), Horizontal axis: 500 µs/div.

**Figure 17 sensors-21-02699-f017:**
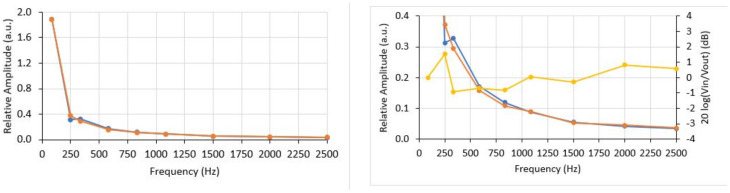
FFT of input (blue line) and output (red line) of the OVT. The graph on the (**right**) is an amplification of the lower portion of the graph on the (**left**). The yellow line is the OVT response in decibel.

**Figure 18 sensors-21-02699-f018:**
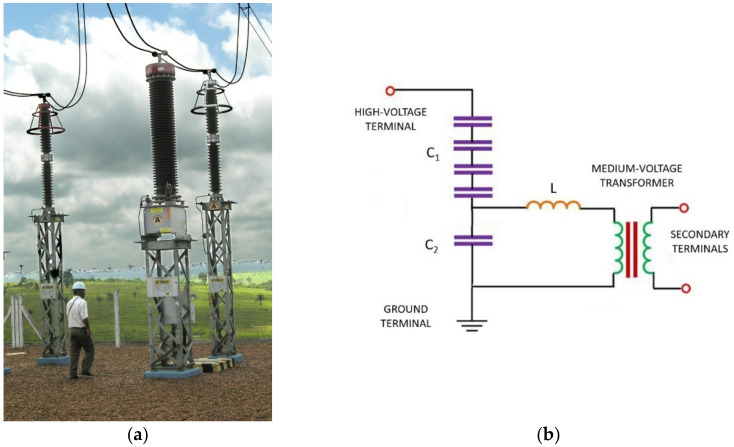
(**a**) A conventional capacitive voltage transformer (CVT) for 230 kV installed on the Altamira Substation in the state of Pará, Brazil. (**b**) Internal circuit diagram showing the capacitive divisor.

**Figure 19 sensors-21-02699-f019:**
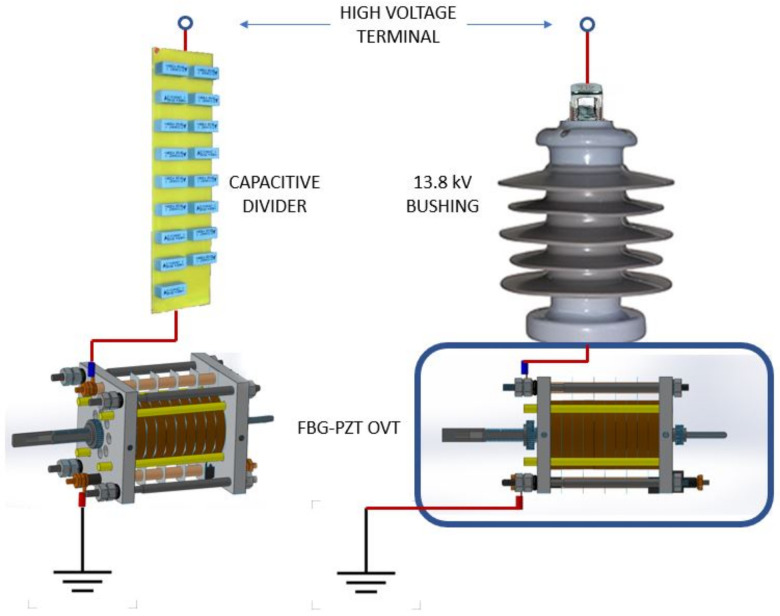
(**Left**) A high voltage capacitor stack was used to lower the applied voltage to the OVT. (**Right**) The proposed assembly to allow the OVT to be used in field measurements with the capacitor stack inserted inside a 13.8 kV bushing, whereas the OVT is protected inside a IP65 casing.

**Table 1 sensors-21-02699-t001:** FBG and PZT Properties.

Parameter	Value
FBG central Bragg wavelength	$\lambda_{B}=1554.220\;\mathrm{nm}$
Silica thermal expansion coefficient	$\alpha_{\mathrm{FBG}}=0.55\times 10^{-6}{{}^{\circ}C}^{-1}$
Thermo-optic coefficient	$\eta =6.541\times 10^{-6}{{}^{\circ}C}^{-1}$
Photoelastic coefficient	$\rho_{e}=0.252$
FBG length	$L_{\mathrm{FBG}}=15\;\mathrm{mm}$
Piezoelectric charge constant (axis 3)	$d_{33}=300\;\mathrm{pm}/V$
PZT width	$w_{o}=8\;\mathrm{mm}$
Internal diameter of PZT ring	d = 38.0 mm
External diameter of PZT ring	D = 50.8 mm
Number of elements in stack	N = 10
Capacitance of the PZT stack	$C_{\mathrm{PZT}}=13.3\;\mathrm{nF}$
Resonance frequency of PZT ring	f_R_ = 186 kHz

**Table 2 sensors-21-02699-t002:** Categories and typical characteristics of power system phenomena according to the IEEE 1159-2019 standard [1].

Categories	Typical Duration
**Transients**	
1. Impulsive	
1.1 Nanosecond	<50 ns
1.2 Microsecond	50 ns–1 ms
1.3 Millisecond	>1 ms
2. Oscillatory	
2.1 Low frequency	0.3–50 ms
2.2 Medium frequency	20 µs
2.3 High frequency	5 µs
**Short-duration rms variations**	
1. Instantaneous	
1.1 ag	0.5–30 cycles
1.2 Swell	0.5–30 cycles
2. Momentary	
2.1 Interruption	0.5 cycles–3 s
2.2 Sag	30 cycles–3 s
2.3 Swell	30 cycles–3 s
3. Temporary	
3.1 Interruption	>3 s–1 min
3.2 Sag	>3 s–1 min
3.3 Swell	>3 s–1 min

**Table 3 sensors-21-02699-t003:** Difference between the THDs of input and output signals for sag, swell and interruption disturbances, representing the distortion introduced by the proposed OVT.

Disturbance	Input THD	Output THD	Difference THD
Sag	5.87%	6.65%	0.77%
Swell	13.85%	15.72%	1.87%
Interruption	9.53%	11.01%	1.48%

## Data Availability

Not applicable.

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
