# Peer review of "Optical Voltage Transformer Based on FBG-PZT for Power Quality Measurement"

_sensors, 2021, doi:10.3390/s21082699_

Round 1

Reviewer 1 Report

(1) In Fig.3, the outer screw is included in the upper series, and the inner screw is included in the lower series.  However, in the decriptions at lines 20-202 on page 5, the opposite arrangement is indicated.  Appropriate corrections are required.

(2) Attachment scheme between the PZT element and the optical fiber is essential for realizing satisfactory performances of the propose device.  However, there are no explanations found in this atricle.  The authors should explain where and how the PZT element and the optical fiber are attached to each other, with appropriate reference to the photo in Fig.2.   

Author Response

Reviewer 1

Comments and Suggestions for Authors

  • In Fig.3, the outer screw is included in the upper series, and the inner screw is included in the lower series.  However, in the decriptions at lines 20-202 on page 5, the opposite arrangement is indicated.  Appropriate corrections are required.

A: Corrected and revised. Please, check in the manuscript; all corrections are in highlighted in yellow

  • Attachment scheme between the PZT element and the optical fiber is essential for realizing satisfactory performances of the propose device.  However, there are no explanations found in this atricle.  The authors should explain where and how the PZT element and the optical fiber are attached to each other, with appropriate reference to the photo in Fig.2.   

A: We improved the explanation that was really confusing. We also indicated in the photo the position of the screw ends.

Reviewer 2 Report

A brief summary

Marceli Gonçalves, Marcelo M. Werneck present an emerging approach of an optical voltage transformer based on fiber Bragg grating (FBG) and piezoelectric ceramics for power quality measurements in 13.8 kV distribution lines. The Paper is interesting, generally well written, and worth publishing. The article may be interesting for many readers. The work presents interesting results of research. However, it requires some improvement. First, the main purpose of the work was not clearly defined enough. The authors should clearly highlight the main aim of the work in the abstract first, and then in the introduction. The novelty of the described research should be emphasized in relation to the work of other authors and, in particular, to the authors' own works [6, 7, 12]. The results of the experiments have not been described in sufficient detail. They are presented too broadly and imprecisely. Numerical values of OVT parameters should be determined, such as sensitivity, errors, uncertainties, frequency response, and others. For example, statements such as "high accuracy", "is capable to reproduce with accuracy", "good reproducibility" are too general, imprecise, and insufficient to consider this publication a scientific article. The research results obtained should be refined and better presented. Conclusions should be more detailed. It is worth extending the discussion of results to scientific aspects. There are also some mistakes in editing the text. These shortcomings significantly reduce the value of the entire article. Therefore I believe that specific changes, additions and revisions are necessary.

Broad comments

  1. Generally, Authors should once again go through MDPI recommendations included in the Instructions for Authors (https://www.mdpi.com/journal/sensors/instructions), sensors-template.dot file (https://www.mdpi.com/files/word-templates/ sensors-template.dot), Manuscript English Editing, Guidelines for Authors (https://www.mdpi.com/authors/english-editing) and make sure they comply with this recommendations. In particular, the authors should pay attention to the following MDPI recommendations:

- In the abstract, the authors should clearly highlight the purpose of the work, and summarize the article's main findings,

- The abstract should be an objective representation of the article: it must not contain results which are not presented and substantiated in the main text,

- Keywords are not capitalized,

- The introduction should define the purpose of the work and its significance, including specific hypotheses being tested. The main aim of the work should be mentioned and the main conclusions highlighted,

- Materials and methods should be described with sufficient detail to allow others to replicate and build on published results,

- A concise and precise description of the experimental results should be provided, their interpretation as well as the experimental conclusions that can be drawn.

- All words in headings should be capitalized,

- Abbreviations should be defined the first time they are mentioned in the abstract, text; also the first time they are mentioned in a table or figure,

- All figures and tables should be cited in the main text as Figure 1, Table 1, etc.,

- A figure caption on a single line should be centered,

- Titles of figures and tables should end with a full stop,

- Figures and tables should be placed in the main text near to the first time they are cited,

- Equations should be punctuated as regular text,

- All symbols representing physical quantities and variables should be italicized,

- All symbols used in the equations, tables, figures and text should be explained,

- The manuscript should not contain any information that has already been published. If you include already published figures or images, please obtain the necessary permission from the copyright holder to publish under the CC-BY license,

- The digital object identifier (DOI) should be included for all references where available.

  1. What is the purpose of this work? This is not clearly stated. First, the authors should clearly highlight the purpose of the work in the abstract. Then, the main aim of the work should be briefly mentioned in the introduction.
  2. The abstract should be an objective representation of the article. It should follow the style of structured abstracts: place the question addressed in a broad context and highlight the purpose of the study, describe briefly the main methods, summarize the article's main findings, and indicate the main conclusions or interpretations. Therefore, the summary should be redrafted, corrected and improved.
  3. The introduction should briefly define the purpose of the work and its significance, including specific hypotheses being tested. The authors should briefly mention the main aim of the work and highlight the main conclusions. Therefore, the introduction needs to be corrected and supplemented.
  4. The results of the research are quite interesting and it seems that the proposed method can be widely used in the future. However, there is no clearly expressed scientific aspect in the work. In its current form, the work looks like a very extended test report. The article should specify the scientific problem that has been solved.
  5. The digital object identifier (DOI) for all references where available should be included. As you can easily check, DOI is available for the vast majority of references. So, it should be improved.
  6. Punctuation in sentences containing equations should be improved. According to the "sensors-template.dot" (see lines 72-75 in this file), equations should be punctuate as regular text. All equations and other mathematical expressions, both in running text and when displayed on separate lines, should be accompanied with appropriate punctuation, according to their function in the sentence. For example, a comma should be written after equation (1) immediately.
  7. Plagiarism is not acceptable. Plagiarism includes copying text, ideas, images, or data from another source, even from your own publications, without giving any credit to the original source. Figures 2, 3, and 9 have already been published in [7]. The prior source of these drawings should be clearly stated and the copyright for their publication should be demonstrated.

Specific comments

Lines 9-21. Abstract.

What is the main aim of this work? This is not clearly stated.In the abstract, the main purpose of the work was not clearly defined enough. The authors should clearly highlight the purpose of the work, and summarize the article's main findings.

Line 22. Keywords are not capitalized.

Lines 25-121. Introduction.

The introduction has been written very well and provides a good background. But, in the introduction, the authors should explain the main aim of the work and finally highlight the main conclusions.

The authors wrote: “In this paper, an emerging approach of an OVT based on fiber Bragg grating (FBG) for power quality measurements is presented. The aim is to provide a signal with an appropriate accuracy required by the power quality analyzers and meters.”

Is the aim of this article to provide a signal with the appropriate accuracy required by power quality analyzers and meters?

It is not well written. It makes no logical sense.

So, the introduction needs to be corrected and supplemented.

Line 20. The authors wrote: “The unique advantage of such system is that it can be monitored up to several kilometers away by a single optical fiber.” Is this really true? The authors did not conduct any research in this direction. On what basis do they make this conclusion? Which device should be used for this purpose? Which optical fiber, what light source, etc.

Line 106. What does PZT abbreviation mean? Abbreviations should be defined the first time they are mentioned in the abstract, text; also the first time they are mentioned in a table or figure

Line 122. Materials and Methods. Materials and methods should be described with sufficient detail to allow others to replicate and build on published results. The authors should provide the types of apparatus used, manufacturers, specification of the main parameters (measuring ranges, accuracy), the type and manufacturer of the optical fiber, Bragg grating, light source, Fabry-Perot filter, PZT rings and other important elements used in the research. Without this data, the article is worthless to the reader.

Line 123. All words in headings should be capitalized.

Lines 133, 144 and others. Equations should be punctuate as regular text. Comma should be written after equation (1) immediately. Likewise in other places.

Line 134 and equation (1). All symbols representing physical quantities and variables should be italicized.

Lines 145, 146 and equation (2). All symbols representing physical quantities and variables should be italicized.

Line 145. The term "longitudinal displacement" is not applied correctly. This is not a displacement, it is a strain. Bragg grating does not displace, Bragg grating is stretched. The word displacement implies that an object has moved, or has been displaced. Displacement is defined to be the change in position of an object.

Line 156. Equations (3), (4). Where does this data come from? Has it been measured by the authors?

Line 159. “ration” ? Maybe ratio or relation?

Line 159 and equation (5). All symbols representing physical quantities and variables should be italicized.

Lines 162, 166. What does w0 symbol mean? “displacement in width” or „thickness of the ring” ? It is not quite the same.

Lines 169, 186 and many, many others. All figures and tables should be cited in the main text as Figure 1, Table 1, etc.

Line 170, Figure 1. Two attachment points should be marked and described in the figure.

Lines 171, 248 and others. A figure caption on a single line should be centered.

Lines 174. "displacement of its thickness" This needs to be corrected, the thickness cannot be displaced. Displacement is defined to be the change in position of an object. In this device, there is a change in thickness, not a displacement.

Lines 192, 206, 317. These drawings were published earlier in [7]. This should be clarified. Is not that plagiarism?

Line 214. Titles of figures and tables should end with a full stop.

Lines 146, 214, Table 1. The coefficients (Silica thermal expansion coefficient, Photoelastic coefficient) have different values. Why? Which are correct? All symbols representing physical quantities and variables should be italicized.

Materials and methods should be described with sufficient detail to allow others to replicate and build on published results. What optical fiber was used? Who is the manufacturer of the Bragg Grating, PZT rings? What kind of glue is used for bonding fibers?

Lines 217, equations (8), (9), (10). All symbols representing physical quantities and variables should be italicized. Equations should be punctuate as regular text.

Line 227. Interrogation System. The components in figure 4 should be described: light source, filter, photodetector, amplifier, oscilloscope. The authors should provide the types of apparatus used, manufacturers, specification of the main parameters.

Line 231. What type of filter was used? Who is the manufacturer?

Lines 233, 246. Figure 4, not Fig. 4.

Line 248. A figure caption on a single line should be centered.

Line 249. Experimental Set-up for Quality Measurement. The components in figure 5 should be described. The authors should provide the types of apparatus used, manufacturers, specification of the main parameters.

Line 250. Figure, not Fig.

Line 295. Figure 8, not Fig. 8. How were these measurements made? There is no appropriate apparatus in Figure 5. Something is missing here, it is unclear.

Line 306. What is the amplification of the amplifier? What type of transformer was used? What is the voltage ratio of this transformer? The manufacturer, type and basic parameters (accuracy, ranges, etc.) of the voltmeter and oscilloscope used in this work should be provided.

Line 311. “Due to the probe output impedance been ten times smaller than the oscilloscope's input impedance, the value displayed is always half the RMS value actually applied to the OVT.”

That is not true! This is inconsistent with the measurement theory. Is the voltmeter measuring correctly?

If the output impedance of the probe is ten times less than the input impedance of the oscilloscope, the oscilloscope measures the voltage correctly. This should be clarified. What are the probe parameters? What are the parameters of the oscilloscope?

Typically, a 1: 1000 high voltage probe has a 1 Giga Ohm resistor and requires the use of a voltmeter (or oscilloscope) with an input resistance of 1 Mega Ohm.

Line 323. “It is possible to notice that the output signal perfectly reproduces the phenomena introduced in the input.” Sorry, but writing like this in a science article is dubious and unbelievable! What does "perfectly reproduces" mean? This should be a measure of power quality, not looking at picture reproductions in a museum.

What accuracy of measurements was obtained? What are the measurement errors? What are the uncertainties? What parameters does the built OTV have? What is his sensitivity? What is the processing constant? What is the nonlinearity? What's the frequency response? This should be calculated and stated in the article. Then it will be the scientific research.

Line344. “The obtained results demonstrate that optical voltage transformer is capable to reproduce with accuracy a short-duration voltage variation applied to its input. The output signal could be directed to a power quality analyzers and meters with good reproducibility.” The same problem. What does that mean in a research article? In my opinion, that means very little, almost nothing.

Line 349. What does “higher frequencies” mean? What frequency value was used?

Line 351. “However, after the signal was directed to the instrument transformer, it was suppressed, since the transformer works with a nominal frequency of 60 Hz and does not respond to high frequencies.” This is an uncertain statement. The 60Hz transformer carries higher frequencies. Figure 15 shows that the transformer carries a voltage of 2 kHz. This should be clarified. You need to determine the parameters of the transformer and estimate its frequency response. They can also be easily measured. The circuit shown in Figure 5 can be used. Without any changes, all you need to do is program the generator. Such measurements can probably be made in an hour, I suppose.

Line 362. I do not understand this measurement. Was DC voltage connected to the transformer? Please explain this in the article.

Line 377. What does "crescent frequency" mean? I am sorry, but I do not know such a term in the measuring technique.

Line 380. This conclusion is questionable. Figure 15 shows that the OVT output is significantly attenuated. The OVT bandwidth should be determined. This is very easy to measure. Authors should make such measurements. The measuring system is ready, the same as shown in Figure 5. Then it will be possible to draw the correct conclusions.

Line415. Conclusions. The conclusions are too general and not very specific. The presented results of measurements are incomplete and require supplementing. Appropriate numerical parameters of the constructed OVT should be calculated and presented. The obtained results of measurements should be better analyzed and commented on. Authors should discuss the results and how they can be interpreted in perspective of previous studies and of the working hypotheses. Future research directions may also be mentioned.

Line 431. This conclusion is not supported by any studies.

Line 440, References. The digital object identifier (DOI) for all references where available should be included. As you can easily check, DOI is available for the vast majority of references. So, it should be improved.

Author Response

Reviewer 2
I would like to thank you for the detailed revision which, we acknowledge, must have taken a lot of time. We tried our best to fulfil all your comments and suggestions, which are highlighted in yellow.  I am sure the manuscript improved a lot after all these revisions.

Broad comments
1.    Generally, Authors should once again go through MDPI recommendations included in the Instructions for Authors (https://www.mdpi.com/journal/sensors/instructions), sensors-template.dot file (https://www.mdpi.com/files/word-templates/ sensors-template.dot), Manuscript English Editing, Guidelines for Authors (https://www.mdpi.com/authors/english-editing) and make sure they comply with this recommendations. In particular, the authors should pay attention to the following MDPI recommendations:
- In the abstract, the authors should clearly highlight the purpose of the work, and summarize the article's main findings,
- The abstract should be an objective representation of the article: it must not contain results which are not presented and substantiated in the main text,
- Keywords are not capitalized,
- The introduction should define the purpose of the work and its significance, including specific hypotheses being tested. The main aim of the work should be mentioned and the main conclusions highlighted,
- Materials and methods should be described with sufficient detail to allow others to replicate and build on published results,
- A concise and precise description of the experimental results should be provided, their interpretation as well as the experimental conclusions that can be drawn.
- All words in headings should be capitalized,
- Abbreviations should be defined the first time they are mentioned in the abstract, text; also the first time they are mentioned in a table or figure,
- All figures and tables should be cited in the main text as Figure 1, Table 1, etc.,
- A figure caption on a single line should be centered,
- Titles of figures and tables should end with a full stop,
- Figures and tables should be placed in the main text near to the first time they are cited,
- Equations should be punctuated as regular text,
- All symbols representing physical quantities and variables should be italicized,
- All symbols used in the equations, tables, figures and text should be explained,
- The manuscript should not contain any information that has already been published. If you include already published figures or images, please obtain the necessary permission from the copyright holder to publish under the CC-BY license,
- The digital object identifier (DOI) should be included for all references where available.
2.    What is the purpose of this work? This is not clearly stated. First, the authors should clearly highlight the purpose of the work in the abstract. Then, the main aim of the work should be briefly mentioned in the introduction.

A02: Abstract and introduction were changed.

3.    The abstract should be an objective representation of the article. It should follow the style of structured abstracts: place the question addressed in a broad context and highlight the purpose of the study, describe briefly the main methods, summarize the article's main findings, and indicate the main conclusions or interpretations. Therefore, the summary should be redrafted, corrected and improved.

A03: Done.

4.    The introduction should briefly define the purpose of the work and its significance, including specific hypotheses being tested. The authors should briefly mention the main aim of the work and highlight the main conclusions. Therefore, the introduction needs to be corrected and supplemented.

A04: Done

5.    The results of the research are quite interesting and it seems that the proposed method can be widely used in the future. However, there is no clearly expressed scientific aspect in the work. In its current form, the work looks like a very extended test report. The article should specify the scientific problem that has been solved.

A05: Done. Please, check abstract, introduction and conclusions.

6.    The digital object identifier (DOI) for all references where available should be included. As you can easily check, DOI is available for the vast majority of references. So, it should be improved.

A06: Done.

7.    Punctuation in sentences containing equations should be improved. According to the "sensors-template.dot" (see lines 72-75 in this file), equations should be punctuate as regular text. All equations and other mathematical expressions, both in running text and when displayed on separate lines, should be accompanied with appropriate punctuation, according to their function in the sentence. For example, a comma should be written after equation (1) immediately.

A07: Done

8.    Plagiarism is not acceptable. Plagiarism includes copying text, ideas, images, or data from another source, even from your own publications, without giving any credit to the original source. Figures 2, 3, and 9 have already been published in [7]. The prior source of these drawings should be clearly stated and the copyright for their publication should be demonstrated.

A08: Permission of Measurement journal (Elsevier) was granted and the source of the figures was informed on the caption.

Specific comments
1)    Lines 9-21. Abstract.
What is the main aim of this work? This is not clearly stated.In the abstract, the main purpose of the work was not clearly defined enough. The authors should clearly highlight the purpose of the work, and summarize the article's main findings.

A: Done. Please, check abstract, introduction and conclusions.

2)    Line 22. Keywords are not capitalized.

A: OK

3)    Lines 25-121. Introduction.
The introduction has been written very well and provides a good background. But, in the introduction, the authors should explain the main aim of the work and finally highlight the main conclusions.
The authors wrote: “In this paper, an emerging approach of an OVT based on fiber Bragg grating (FBG) for power quality measurements is presented. The aim is to provide a signal with an appropriate accuracy required by the power quality analyzers and meters.”
Is the aim of this article to provide a signal with the appropriate accuracy required by power quality analyzers and meters?
It is not well written. It makes no logical sense.
So, the introduction needs to be corrected and supplemented.

A: Introduction was changed accordingly.

4)    Line 20. The authors wrote: “The unique advantage of such system is that it can be monitored up to several kilometers away by a single optical fiber.” Is this really true? The authors did not conduct any research in this direction. On what basis do they make this conclusion? Which device should be used for this purpose? Which optical fiber, what light source, etc.

A: An explanation and a reference has been added.

5)    Line 106. What does PZT abbreviation mean? Abbreviations should be defined the first time they are mentioned in the abstract, text; also the first time they are mentioned in a table or figure

A: Done

6)    Line 122. Materials and Methods. Materials and methods should be described with sufficient detail to allow others to replicate and build on published results. The authors should provide the types of apparatus used, manufacturers, specification of the main parameters (measuring ranges, accuracy), the type and manufacturer of the optical fiber, Bragg grating, light source, Fabry-Perot filter, PZT rings and other important elements used in the research. Without this data, the article is worthless to the reader.

A: OK. Section Materials and Methods now contains specifications of all equipment used.

7)    Line 123. All words in headings should be capitalized.
A: Done.

8)    Lines 133, 144 and others. Equations should be punctuate as regular text. Comma should be written after equation (1) immediately. Likewise in other places.

A: Done.

9)    Line 134 and equation (1). All symbols representing physical quantities and variables should be italicized.

A: Done.

10)    Lines 145, 146 and equation (2). All symbols representing physical quantities and variables should be italicized.

A: Done.

11)    Line 145. The term "longitudinal displacement" is not applied correctly. This is not a displacement, it is a strain. Bragg grating does not displace, Bragg grating is stretched. The word displacement implies that an object has moved, or has been displaced. Displacement is defined to be the change in position of an object.

A: OK, changed.

12)    Line 156. Equations (3), (4). Where does this data come from? Has it been measured by the authors?

A: By substituting data from Table 1 in Eq. (2) we get to equations (3) and (4). Those are theoretical values, but by calibrating the FBG to strain and temperature, we get to approximately the same values.

13)    Line 159. “ration” ? Maybe ratio or relation?

A: relation

14)    Line 159 and equation (5). All symbols representing physical quantities and variables should be italicized.

A: OK, done.

15)    Lines 162, 166. What does w0 symbol mean? “displacement in width” or „thickness of the ring” ? It is not quite the same.

A: The text was improved.

16)    Lines 169, 186 and many, many others. All figures and tables should be cited in the main text as Figure 1, Table 1, etc.

A: OK, done.

17)    Line 170, Figure 1. Two attachment points should be marked and described in the figure.

A: The figure was improved and a text has been added.

18)    Lines 171, 248 and others. A figure caption on a single line should be centered.

A: OK, done.

19)    Lines 174. "displacement of its thickness" This needs to be corrected, the thickness cannot be displaced. Displacement is defined to be the change in position of an object. In this device, there is a change in thickness, not a displacement.

A: OK, corrected.

20)    Lines 192, 206, 317. These drawings were published earlier in [7]. This should be clarified. Is not that plagiarism?

A: Permission of Measurement journal (Elsevier) was granted and the source of the figures was informed on the caption.

21)    Line 214. Titles of figures and tables should end with a full stop.

A: OK, done.

22)    Lines 146, 214, Table 1. The coefficients (Silica thermal expansion coefficient, Photoelastic coefficient) have different values. Why? Which are correct? All symbols representing physical quantities and variables should be italicized.

A: They are different parameters. Photoelastic coefficient is the sensitivity of the refractive index of silica with strain.

23)    Materials and methods should be described with sufficient detail to allow others to replicate and build on published results. What optical fiber was used? Who is the manufacturer of the Bragg Grating, PZT rings? What kind of glue is used for bonding fibers?

A: OK. Materials and Methods now contains specifications of all equipment used.

24)    Lines 217, equations (8), (9), (10). All symbols representing physical quantities and variables should be italicized. Equations should be punctuate as regular text.

A: OK, done.

25)    Line 227. Interrogation System. The components in figure 4 should be described: light source, filter, photodetector, amplifier, oscilloscope. The authors should provide the types of apparatus used, manufacturers, specification of the main parameters.

A: All components shown in the interrogation system of Fig. 4 have been described and specified.

26)    Line 231. What type of filter was used? Who is the manufacturer?

A: Informed on Section 2.3 Interrogation System

27)    Lines 233, 246. Figure 4, not Fig. 4.

A: OK, done.

28)    Line 248. A figure caption on a single line should be centered.

A: OK, done.

29)    Line 249. Experimental Set-up for Quality Measurement. The components in figure 5 should be described. The authors should provide the types of apparatus used, manufacturers, specification of the main parameters.

A: All components shown in the experimental setup of Fig. 5 have been described and specified

30)    Line 250. Figure, not Fig.

A: OK, done.

31)    Line 295. Figure 8, not Fig. 8. How were these measurements made? There is no appropriate apparatus in Figure 5. Something is missing here, it is unclear.

A: All the patterns shown in Fig. 8 were repetitive. To measure them we used the same oscilloscope shown in Fig. 5, firstly measuring input and output of the amplifier and then input and output of the OVT. It would pollute too much Fig 5 if we included another oscilloscope in the setup.

32)    Line 306. What is the amplification of the amplifier? What type of transformer was used? What is the voltage ratio of this transformer? The manufacturer, type and basic parameters (accuracy, ranges, etc.) of the voltmeter and oscilloscope used in this work should be provided.

A: The basic parameters were informed when these instruments were first mentioned.

33)    Line 311. “Due to the probe output impedance been ten times smaller than the oscilloscope's input impedance, the value displayed is always half the RMS value actually applied to the OVT.”
That is not true! This is inconsistent with the measurement theory. Is the voltmeter measuring correctly?
If the output impedance of the probe is ten times less than the input impedance of the oscilloscope, the oscilloscope measures the voltage correctly. This should be clarified. What are the probe parameters? What are the parameters of the oscilloscope?
Typically, a 1: 1000 high voltage probe has a 1 Giga Ohm resistor and requires the use of a voltmeter (or oscilloscope) with an input resistance of 1 Mega Ohm.

A: The text was really wrong and it has been corrected. The high voltage probe output impedance is 1.1 Mega Ohm (Manufacture Minipa, model HV-40A) and requires a voltmeter with input impedance of 10 Mega Ohm. However, the oscilloscope used has an input impedance equal to 1 Mega Ohm (Manufacture Rigol, model DS1102CA). That’s why the value displayed is about half of the real value.
A description of each equipment was included in the text.

34)    Line 323. “It is possible to notice that the output signal perfectly reproduces the phenomena introduced in the input.” Sorry, but writing like this in a science article is dubious and unbelievable! What does "perfectly reproduces" mean? This should be a measure of power quality, not looking at picture reproductions in a museum.

A: The term "perfectly reproduces" was removed. The theorical output/input ratio was calculated in Eq. (14) which was used to compare the adequate reproduction of the input signal by the OVT. Errors were calculated in each case and informed in the text. Another method was used to check the adequate reproducibility of the OVT which consists on measuring the THD of the output and compare it with the THD of the input. For this calculation it was necessary to take the Fast Fourier Transform (FFT) of both input and output of the three disturbances as shown in the figures that follow the disturbances. The THDs are shown in Table 3.
See also the answer of Question 41.

35)    What accuracy of measurements was obtained? What are the measurement errors? What are the uncertainties? What parameters does the built OTV have? What is his sensitivity? What is the processing constant? What is the nonlinearity? What's the frequency response? This should be calculated and stated in the article. Then it will be the scientific research.

A: All parameters suggested above were included in the text. As for the frequency response of the OVT, one can mention that the PZT ring possess a resonance frequency of 186 kHz (see Table 1). This means that it can oscillate at least up this frequency, which encompass all important harmonics when one wants to investigate THD or high-frequency surges.
Of course, one has to take into account the inertia offered by the movable aluminum plate (see Fig. 2), and the high Young modulus of the optical fiber that has to be stretched; both will interfere in the frequency response. We did not include this discussion in the text because, instead of being clarifying, it is just an estimation and it could mislead and confuse the reader.
Conventional procedures to calculate the frequency response of the OVT did not apply in our case because it was not possible to reproduce either high-frequency disturbances by the instrument transformer such as transient and step, or high frequency sinusoidal waveshapes (see answer of Question 39). For this reason, we applied another approach to know which frequencies can be transmitted by the OVT. The approach is to calculate the Fast Fourier Transform (FFT) of an input signal applied to the OVT and compare it with the FFT of its corresponding response. Then, for each frequency pair (input and output), we can compare their amplitudes to check when the attenuation reaches the -3dB cut-off frequency. Figure 17 shows the two FFTs taken from the signals shown in Figure 15b: the input and output of the OVT.
Figure 17(left) shows input FFT superimposed with the output FFT for frequencies up to 2.5 kHz. Notice that they are very close to each other, but in order to reveal small differences, Figure 17(right) presents an amplified portion of the superimposed FFTs. It is possible to see that in some frequencies the OVT response slightly attenuates the input signal and in other frequencies the opposite happens. The graph in Figure 17(right) also shows the OVT’s response in dB (yellow line). In all frequencies up to 2.5 kHz, the response does not reach the  3 dB point, which means that the cut-off frequency was not reached. The conclusion is that, the cut-off frequency was not detected in this range and the OVT is capable to reproduce at least up to the 41st harmonic without distortion.

36)    Line344. “The obtained results demonstrate that optical voltage transformer is capable to reproduce with accuracy a short-duration voltage variation applied to its input. The output signal could be directed to a power quality analyzers and meters with good reproducibility.” The same problem. What does that mean in a research article? In my opinion, that means very little, almost nothing.

A: See the answer of Question 34.

37)    Line 349. What does “higher frequencies” mean? What frequency value was used?

A: OK, “higher frequencies” was not clear. Changed to: “transients (see Table 2)”.

38)    Line 351. “However, after the signal was directed to the instrument transformer, it was suppressed, since the transformer works with a nominal frequency of 60 Hz and does not respond to high frequencies.” This is an uncertain statement. The 60Hz transformer carries higher frequencies. Figure 15 shows that the transformer carries a voltage of 2 kHz. This should be clarified. You need to determine the parameters of the transformer and estimate its frequency response. They can also be easily measured. The circuit shown in Figure 5 can be used. Without any changes, all you need to do is program the generator. Such measurements can probably be made in an hour, I suppose.

A: Yes, the transformer can transmit frequencies higher than 60 Hz, for example 1 kHz as shown in Fig. 15 (now 16). Notice that the horizonal axis scale is 500 s/DIV. However, the transformer functions as a low-pass filter, attenuating frequencies higher than its 3dB cut-off frequency. To find its cut-off frequency, we start by noticing that in Fig. 14a (now 15a) a step function was applied on the transformer, (CH2-red line). The transformer responded with a typical attenuated step function shown in Fig. 14b (now 15b) (blue line), whose rise-time indicates the f(3dB) frequency: RISE_TIME=0.35/f(3dB). This relationship is valid for many first-order, electrical and electro-optical systems such as photodiode-based systems. Then, from Fig. 14(b), we calculated the rise time (time between 10% and 90% of the response) = 5.2 ms. Then, by the equation above, f(3dB) = 66.8 Hz. Therefore, the cut-off frequency response of the transformer is 67.3 Hz, or, in another words, any input frequency above this value will be attenuated at about  20dB/decade. And this is the reason for we could not apply either a step function or several sinusoidal waveshapes on the OVT to calculate its frequency response.
Another method to calculate the transformer cut-off frequency is by comparing the FFT of both its input and output. We did this and noticed that the output decay of the FFT components reach 0.707 at about the same frequency as calculated above, which confirms the calculation described above.
We included just a shrank version of this explanation in the text, because the purpose of the paper was not intended to study the transformer. 

39)    Line 362. I do not understand this measurement. Was DC voltage connected to the transformer? Please explain this in the article.

A: This measurement was performed by simulating a step in the Matlab and injecting it in the amplifier to check the step response of the transformer and the OVT. This explanation was given in the text.

40)    Line 377. What does "crescent frequency" mean? I am sorry, but I do not know such a term in the measuring technique.

A: The term "crescent frequency" was changed

41)    Line 380. This conclusion is questionable. Figure 15 shows that the OVT output is significantly attenuated. The OVT bandwidth should be determined. This is very easy to measure. Authors should make such measurements. The measuring system is ready, the same as shown in Figure 5. Then it will be possible to draw the correct conclusions.

A: Apart from using a step function to calculate de frequency response of an electronic system we can also use the conventional technique which is by increasing the frequency of a pure sinusoidal signal and observing the response. When the output signal decays by 0.707, the 3dB point was reached. However, as explained on Question 38, it was not possible to apply a step function due to the limited frequency response of the transformer. The other method was also unfruitful, as since we increased the input frequency, the output decreases sharply. For 100 Hz the attenuation was about 30%, at 200 Hz the output signal was already attenuated by half the value of the input and so on. This explains the limited amplitude reached on the test of Fig. 15 (now 16), in which the frequency is 1 kHz, but at this frequency the attenuation is so high that we could not reach the 4,000 V, instead, we reached only 536 V, as shown in Fig 15(now 16) (blue line) (*).
See also the answer of Question 34.
(*) The real value measured by the oscilloscope with the 1000X probe is the double to the values shown, multiplied by 1,000 (see Question 33)

42)    Line415. Conclusions. The conclusions are too general and not very specific. The presented results of measurements are incomplete and require supplementing. Appropriate numerical parameters of the constructed OVT should be calculated and presented. The obtained results of measurements should be better analyzed and commented on. Authors should discuss the results and how they can be interpreted in perspective of previous studies and of the working hypotheses. Future research directions may also be mentioned.

A: Section CONCLUSIONS was improved.

43)    Line 431. This conclusion is not supported by any studies.

A: This statement was inserted in the INTRODUCTION and appropriated reference was informed.

44)    Line 440, References. The digital object identifier (DOI) for all references where available should be included. As you can easily check, DOI is available for the vast majority of references. So, it should be improved.

A: Done 

Reviewer 3 Report

please find my comments in the report attached.

Author Response

Reviewer 3
In this paper, the Authors propose an optical voltage transformer (OVT) based on a fiber Bragg grating (FBG) and piezoelectric ceramics (PZT) for power quality measurements. The OVT proposed was tested under different disturbances according to the standard IEEE 1159-2019. The Authors focused on shortduration voltage variations because they are the most common disturbances encountered. It was found that the OVT was capable to reproduce the voltage of sag, swell and interruption disturbances with high accuracy. The OVT was also demonstrated to be capable of measuring transient disturbances. I find the paper interesting, and I consider it can be published provided that the following issues are addressed in detail. 
1)    Fig. 1-2 can be misleading. Around Fig. 1, it is explained that the FBG is fixed on the PZT ring, but that is not the case in Fig. 2, where it is indicated that the FBG is bonded to screws, to which the strain is transferred from the PZT rings. Please adjust Fig. 1 to correspond to the actual experimental arrangement of Fig. 2. Also, make clearer that Fig. 1 is mainly to illustrate the formalism of the principle of operation, but it may not accurately represent the experimental conditions. 

A: To clarify the understanding, Figure 1 was replaced by a new figure showing the real system with the FBG bonded between the screws, according to the prototype shown in Figure 2. A better explanation was also introduced that makes it clear the strategic used.

2)    Around Fig. 4, is it explained that frequency-modulated signal is converted into an amplitude-modulated one, by using an edge filter. Therefore, the key element in the interrogation system is the Fabry-Perot filter. Please extend the explanation of how this frequency-to-amplitude is carried out by this device. 

A: Figure 4 was improved showing the spectra of the signals as they flow through the different components of the setup. We also included an explanation of the whole process that makes it clear the convolution process applied.

3)    Besides a step-by-step detailed explanation, I strongly suggest including a schematic of how the optical signal e.g., spectrum, looks like at each stage of Fig. 4. For instance, the native spectrum is that of the broadband light source; this spectrum enters the FBG and a narrow-band spectrum is reflected; then, this narrow spectrum enters the FP, and so on. 

A: Figure 4 was changed and a representation of the optical signal for each stage was included. The text was also improved, including a step-by-step explanation.

4)    The results shown in Fig. 8, 10, 11, 12, 13, 14, 15 show that the electrical signal is followed by the OVT. However, the Authors claim that this is performed with “high accuracy”. For these claims of high accuracy to be valid, the Authors must include quantitative metrics of comparison between the two signals e.g., cross-correlation coefficient. 

A: The theorical output/input ratio of the OVT was calculated in Eq. (14) which was used to compare the adequate reproduction of the input signal by the OVT. Errors were calculated in each case and informed in the text. Another method was used to check the adequate reproducibility of the OVT which consists on calculating the THD of the output and compare it with the THD of the input. For this calculation it was necessary to take the Fast Fourier Transform (FFT) of both input and output of the three disturbances as shown in the figures that follow the disturbances. The THDs are shown in Table 3.

5)    In Eq. (10) it is stablished that the sensitivity is of 240 pm/kV. This value is rather low and makes the OVT proposed useful mainly for high-voltage machinery. Please explain how this sensitivity can be enhanced for the OVT to be useful also for lower-voltage machinery.

A: Better sensitivities can be achieved by choosing a different PZT type with a higher piezoelectric charge constant, for instance, PZT-5A or PZT-5H from the same manufacturer. The number of elements in the PZT stack also influences the increased sensitivity, however, should be increased with caution, since the assembly could become heavy and bulky. Another way of enhancing the sensitivity is to decrease the FBG length, and reducing the distance between the end of inner and outer screws, where the FBG is bonded, since this would increase the strain applied to the FBG.
This text with examples was included after Eq. (10).

6)     I strongly suggest the Authors to seek help of an English native speaker since the language needs thorough improvement. I found many typos throughout the paper as well as several instances where the content can be made much clearer by writing it differently. Examples of the many instances are the following: Lines 49-50: “For measurement purpose, several power quality monitoring devices are available in the market” Lines 53-55: “Divided into voltage transformers (VT) and current transformers (CT,) they are responsible for converting high voltage or current into standardized values of voltage or current that can be dealt with.” Lines 55-57: “However, for high voltage applications, voltage and current transformers when inserted into the measurement chain could impact on the measurement” reads much better if written as: “However, for high voltage applications, VTs and CTs could impact on the measurement when inserted into the measurement chain”. Lines 219-220: “Substituting Equation (8) into Equation (2), considering that the temperature is fully compensated, we obtain the sensitivity of the OVT under voltage variation” Lines 158-160: “The piezoelectric charge constant (dij) informs what is the ratio between the strain (ε) of the piezoelectric material and the electric field (E) applied to this material”

A: The text, in general, was improved and the highlighted phases were revised.

Reviewer 4 Report

The work by Marceli et al. proposed an optical voltage transformer based on FBG-PZT for power quality measurement. The work is interesting and important for the application of fibre sensors in high voltage sensing. But there remain many issues. I can’t recommend the acceptance till all my concerns have been well responded:

1. The motivation is not clear. As mentioned by the authors that there have some optical transformers based on Pockels and Faraday effects, FBG and Power over Fiber (PoF) techniques. But the key features and difference are not that clear. Especially, the authors emphasize the importance of the behaviour for the disturbance in your work. So how about others? In addition, it is too long for your introduction. If possible, please be more concise and clearer.

2. What is the responsive time of PZT used? Will it be fast enough to follow the fast change?

3. Please use other symbol for the number of ceramic rings, as n is often for refractive index for the optical work.

4. The paper gave out many results. But all are qualitative. You should have or define some assessment factors to quantify your work. How good or difference with others?

5. The English is very casual at many places. The authors should further check and improve it.

Author Response

Reviewer 4
Comments and Suggestions for Authors
The work by Marceli et al. proposed an optical voltage transformer based on FBG-PZT for power quality measurement. The work is interesting and important for the application of fibre sensors in high voltage sensing. But there remain many issues. I can’t recommend the acceptance till all my concerns have been well responded:

1.    The motivation is not clear. As mentioned by the authors that there have some optical transformers based on Pockels and Faraday effects, FBG and Power over Fiber (PoF) techniques. But the key features and difference are not that clear. Especially, the authors emphasize the importance of the behaviour for the disturbance in your work. So how about others? In addition, it is too long for your introduction. If possible, please be more concise and clearer.

A: We improved the introduction mentioning why Pockels and Faraday are mainly avoided by power station operators and the motivation toward alternative optical technologies. The introduction is clearer now, but we could not reduce it too much because other reviewers asked for more information to be placed there.

2.    What is the responsive time of PZT used? Will it be fast enough to follow the fast change?

A: About the frequency response of the OVT, one can mention that the PZT ring possess a resonance frequency of 186 kHz (see Table 1). This means that it can oscillate at least up this frequency, which encompass all important harmonics when one wants to investigate THD or high-frequency surges.
Of course, one has to take into account the inertia offered by the movable aluminum plate (see Fig. 2), and the high Young modulus of the optical fiber that has to be stretched; both will interfere in the frequency response of the system. 
Conventional procedures to calculate the frequency response of the OVT did not apply in our case because it was not possible to reproduce either high-frequency disturbances by the instrument transformer such as transient and step, or high frequency sinusoidal waveshapes. The reason for this is clarified in Fig. 15 showing a step function applied to the instrument transformer and its respective response, an attenuated step, typical of low pass systems such as an inductive transformer. We included in the text the calculation of the frequency response of the transformer, 67.3 Hz. 
For this reason, we applied another approach to know which frequencies can be transmitted by the OVT. The approach is to calculate the Fast Fourier Transform (FFT) of an input signal applied to the OVT and compare it with the FFT of its corresponding response. Then, for each frequency pair (input and output), we can compare their amplitudes to check when the attenuation reaches the -3dB cut-off frequency. Figure 17 shows the two FFTs taken from the signals shown in Figure 15b: the input and output of the OVT.
Figure 17(left) shows input FFT superimposed with the output FFT for frequencies up to 2.5 kHz. Notice that they are very close to each other, but in order to reveal small differences, Figure 17(right) presents an amplified portion of the superimposed FFTs. It is possible to see that in some frequencies the OVT response slightly attenuates the input signal and in other frequencies the opposite happens. The graph in Figure 17(right) also shows the OVT’s response in dB (yellow line). In all frequencies up to 2.5 kHz, the response does not reach the  3 dB point, which means that the cut-off frequency was not reached. The conclusion is that, the cut-off frequency was not detected in this range and the OVT is capable to reproduce at least up to the 41st harmonic without distortion.

3.    Please use other symbol for the number of ceramic rings, as n is often for refractive index for the optical work.

A: OK, done.

4.    The paper gave out many results. But all are qualitative. You should have or define some assessment factors to quantify your work. How good or difference with others?

A: The theorical output/input ratio of the OVT was calculated in Eq. (14) which was used to compare the adequate reproduction of the input signal by the OVT. Errors were calculated in each case and informed in the text. Another method was used to check the adequate reproducibility of the OVT which consists on calculating the THD of the output and compare it with the THD of the input. For this calculation it was necessary to take the Fast Fourier Transform (FFT) of both input and output of the three disturbances as shown in the figures that follow the disturbances. The THDs are shown in Table 3.

5.    The English is very casual at many places. The authors should further check and improve it.

A: The text was thoroughly reviewed and improved.

Reviewer 5 Report

I read the manuscript carefully, it is devoted to a rather interesting research topic related to voltage measurement in electrical conductors. The authors provided a sufficient overview and took a good approach to solve the task. The authors paid special attention to such an important parameter as the compensation of the effect of temperature on the readings of the FBG sensor. The material presented in the manuscript confirms the research carried out by our team as well. Moreover, the authors of the manuscript managed to develop a prototype design. They went further than us.
The manuscript is well-read, well organized, amply illustrated, and the experimental work amply supported by mathematical formulas. I recommend publishing the manuscript as it stands.

Author Response

Reviewer 5

Comments and Suggestions for Authors

I read the manuscript carefully, it is devoted to a rather interesting research topic related to voltage measurement in electrical conductors. The authors provided a sufficient overview and took a good approach to solve the task. The authors paid special attention to such an important parameter as the compensation of the effect of temperature on the readings of the FBG sensor. The material presented in the manuscript confirms the research carried out by our team as well. Moreover, the authors of the manuscript managed to develop a prototype design. They went further than us.
The manuscript is well-read, well organized, amply illustrated, and the experimental work amply supported by mathematical formulas. I recommend publishing the manuscript as it stands.

A: The authors thank you for the constructive comments.

Reviewer 6 Report

Interesting topic, but the paper shoudl be rejected with suggestion of future resubmission.

Make more deep search in literature.
Compare your suggested solution to existing ones.
Compare the simulation results with reality 
Analyze the achievable accuracy and compare with experiments. 
Discuss the effect of temperature variations.

Author Response

Reviewer 6

Comments and Suggestions for Authors

Interesting topic, but the paper shoudl be rejected with suggestion of future resubmission.

Make more deep search in literature.
Compare your suggested solution to existing ones.
Compare the simulation results with reality 
Analyze the achievable accuracy and compare with experiments. 
Discuss the effect of temperature variations.

A: The authors thank you for the suggestions. All comments have been taken care of since most of them were also suggested by the other five reviewers. Please, take a look at the revised manuscript. All changes are in yellow.

Round 2

Reviewer 1 Report

Temperature compensation is a key factor for achieving precise measurement results.  Even if the authors believe that it was decribed in another article (Reference [7]) and is not the main subject of this article, sufficient clear descriptions should be provided for the purpose of realizing better understandings of readers.  From such points of view, the descriptions currently found in the article can be revised further, at least, in view of the following points. 

(1) On page 5, line 216, "two aluminum sustention plate;" may be rewritten as "two aluminum sustention plate (one is fixed, while the other is movable);".

(2) On page 6, line 235, "the movable plate" may be changed as "the movable sustention plate".

(3) On page 6, line 236, "the fixed plate" may be changed as "the fixed sustention plate".

(4) On page 6, line 237, "(see Figure 2)" may be changed as "(see Figure 1)", because the FBG attachment is more clearly illustrated in Fig.1 than in Fig.2.

(5) On page 6, line 238, the description "the two plates (fixed and movable) tend to move together" is confusing, because if the fixed plate can move, it is NOT fixed.  Corrections are required.

(6) The diagram illustrated in Fig.3 is confusing for the following reasons.

6-a)  Both the PZT stack and the inner screw are attached onto the fixed plate, and thus, it is strange for the fixed plated to be included only in the upper series.  Rather, the wall-like stuff at the most-left end looks like the fixed plate.

6-b) At lines 234-235, the lower series is made of the inner screw.  However, the lower series illustrated in Fig.3 includes both the inner screw and the FBG.  In view of the arrangement illustrated in Fig.1, the FBG in Fig.3 should not be illustrated along the same straight line as the inner screw, but should be illustrated as the third line different from but parallel to both of the upper series and the inner screw.

Author Response

Reviewer 1

Temperature compensation is a key factor for achieving precise measurement results.  Even if the authors believe that it was decribed in another article (Reference [7]) and is not the main subject of this article, sufficient clear descriptions should be provided for the purpose of realizing better understandings of readers.  From such points of view, the descriptions currently found in the article can be revised further, at least, in view of the following points. 

(1) On page 5, line 216, "two aluminum sustention plate;" may be rewritten as "two aluminum sustention plate (one is fixed, while the other is movable);".

Answer: The text was changed as suggested.

(2) On page 6, line 235, "the movable plate" may be changed as "the movable sustention plate".

Answer: The text was changed as suggested.

(3) On page 6, line 236, "the fixed plate" may be changed as "the fixed sustention plate".

Answer: The text was changed as suggested.

4) On page 6, line 237, "(see Figure 2)" may be changed as "(see Figure 1)", because the FBG attachment is more clearly illustrated in Fig.1 than in Fig.2.

Answer: The reference was redirected to Figure 1.

(5) On page 6, line 238, the description "the two plates (fixed and movable) tend to move together" is confusing, because if the fixed plate can move, it is NOT fixed.  Corrections are required.

Answer: The text was changed to clarify the understanding.

(6) The diagram illustrated in Fig.3 is confusing for the following reasons.

6-a)  Both the PZT stack and the inner screw are attached onto the fixed plate, and thus, it is strange for the fixed plated to be included only in the upper series.  Rather, the wall-like stuff at the most-left end looks like the fixed plate.

Answer: The wall-like stuff is in reality, a fixed reference, not the fixed plate. The line at the right of the illustration means that the right side of the FBG is fixed to the tip of the outer screw.

The fixed plate is represented by the small double arrow that is touching the reference. We defined the reference to be the left side of the fixed plate, that is, its external side. That is why we call it a “fixed” plate. And since the inner screw is fixed at this reference, it is supposed not to move when this plate expands. That is why the fixed plate appears only on the upper series of the schematic of Fig. 3. The reference could be placed anywhere, but we believed it would be easier to define it there.

When the fixed plate expands with temperature, it will push to the right of the drawing the PZT stack, electrodes, movable plate, and outer screw. The inner screw will not move with respect to the reference due to the fixed plate expansion.

We improved the schematic in Fig. 3 and in the text we also improved the explanation of the schematic.

6-b) At lines 234-235, the lower series is made of the inner screw.  However, the lower series illustrated in Fig.3 includes both the inner screw and the FBG.  In view of the arrangement illustrated in Fig.1, the FBG in Fig.3 should not be illustrated along the same straight line as the inner screw, but should be illustrated as the third line different from but parallel to both of the upper series and the inner screw.

Answer: The FBG lies fixed between the two screw ends. The lower series is made of the inner screw and the FBG bonded to its end. The text and Fig 3 were changed to allow a better understanding.

Reviewer 4 Report

Most of my concerns have well been responded and I recommended the acceptance for the publication, although the introduction remains too long.

Author Response

Most of my concerns have well been responded and I recommended the acceptance for the publication, although the introduction remains too long.

Answer: The authors thank you for the recommendation. Unfortunately, we could not reduce more the introduction because other reviewers asked for more information.

Reviewer 6 Report

All my comments were addressed

Author Response

Reviewer 6

All my comments were addressed

Answer: The authors thank your new review.